# A nucleotide resolution map of Top2-linked DNA breaks in the yeast and human genome

William H. Gittens[1,2]*, Dominic J. Johnson[1,2], Rachal M. Allison [1], Tim J. Cooper[1], Holly Thomas[1] & Matthew J. Neale [1]*

DNA topoisomerases are required to resolve DNA topological stress. Despite this essential role, abortive topoisomerase activity generates aberrant protein-linked DNA breaks, jeopardising genome stability. Here, to understand the genomic distribution and mechanisms underpinning topoisomerase-induced DNA breaks, we map Top2 DNA cleavage with strand-specific nucleotide resolution across the *S. cerevisiae* and human genomes—and use the meiotic Spo11 protein to validate the broad applicability of this method to explore the role of diverse topoisomerase family members. Our data characterises Mre11-dependent repair in yeast and defines two strikingly different fractions of Top2 activity in humans: tightly localised CTCF-proximal, and broadly distributed transcription-proximal, the latter correlated with gene length and expression. Moreover, single nucleotide accuracy reveals the influence primary DNA sequence has upon Top2 cleavage—distinguishing sites likely to form canonical DNA double-strand breaks (DSBs) from those predisposed to form strand-biased DNA single-strand breaks (SSBs) induced by etoposide (VP16) in vivo.

[1] Genome Damage and Stability Centre, University of Sussex, Brighton BN1 9RQ, UK. [2] These authors contributed equally: William H. Gittens, Dominic J. Johnson. *email: w.gittens@sussex.ac.uk; m.neale@sussex.ac.uk

D NA topoisomerases are a broad and ubiquitous family of enzymes that tackle topological constraints to replication, transcription, the maintenance of genome structure and chromosome segregation in mitosis and meiosis[1]. Although the specific mechanisms by which this is accomplished vary considerably across the family, key aspects are shared: including single or double-strand DNA cleavage to form a transient covalent complex (CC), which allows alteration of the topology of the nucleic acid substrate prior to religation[2]. These processes are essential but carry with them a significant risk to genome stability because the CC may be stabilised as a permanent protein-linked DNA break by several physiological factors; such as the proximity of other DNA lesions, the collision of transcription and replication complexes, denaturation of the topoisomerase, or by the binding of small molecules that inhibit religation[1,3]. Topoisomerase-induced DNA strand breaks have been proposed to constitute a significant fraction of the total damage to genomic DNA per day, and have been linked to the genesis and development of various cancers[4,5], including a subset of therapy-related acute myeloid leukaemias (t-AML) caused by the use of Topoisomerase 2 (Top2) poisons in the chemotherapeutic treatment of primary cancers[6,7].

In *S. cerevisiae*, either Top1 or Top2 activity is sufficient to support transcription[8]. However, whilst *top1Δ* cells are viable, Top2 is essential for sister chromatid segregation[9–11]. In contrast to *S. cerevisiae*, all known vertebrate species encode two Top2 proteins (TOP2α and TOP2β). Interestingly, whilst TOP2α is essential for cellular proliferation—cells arrest in mitosis in its absence[12]—TOP2β is not. Rather, TOP2β apparently plays an important role in promoting transcriptional programmes associated with neuronal development[13,14], a function that cannot be supported by TOP2α.

Accurate maps of the positions of Top2-DNA covalent complexes (Top2 CCs) genome-wide and throughout the cell cycle provide insights into topological genome structural organisation, as well as providing the tools to study their repair should they become permanent DNA lesions. Whilst the dual catalytic sites present within the homodimeric Top2 enzyme suggest a primary role in DNA double-strand break (DSB) formation driven by a biological need for decatenation[1], previous research has suggested that etoposide (VP16) and other Top2 poisons may induce a population of DNA single-strand breaks (SSBs), due to independent inhibition of each active site[15–17]. Yet, despite the use of VP16 in chemotherapy, and the different toxicity of these two classes of DNA lesion, the prevalence of Top2-linked SSBs remains unclear.

More recently, general DSB mapping techniques have been applied to map VP16-induced lesions[18–20]. However, such methods are only able to map DSB ends, which therefore excludes VP16-induced SSBs. Moreover, these methodologies lack specificity for protein-linked DNA ends, and require nucleolytic processing to blunt DNA termini as part of sample preparation[18–20]. When combined, these factors lead to loss of nucleotide resolution at the site of Top2 cleavage.

To generate a more complete, and direct, picture of topoisomerase action across the genome, we present here a technique for nucleotide resolution mapping of protein-linked DNA breaks (referred to collectively here as "Covalent Complexes"; CC), which we demonstrate to be generally applicable in both yeast and human cell systems. We establish that the technique is able to map both the positions of Top2 CCs and also CCs of DNA linked to the meiotic recombination protein Spo11, which is related to the archaeal Topoisomerase VI[21]. Furthermore, we provide insights into the spatial distribution of human TOP2 CCs, including comparative analysis of their local enrichment around transcription start sites (TSSs) and CTCF-binding motifs. We

find that TSS-proximal TOP2 CC levels are correlated with transcription in human cells, a question that has come under scrutiny recently[18]. Finally, we present evidence that at some genomic sites VP16 induces a majority of Top2-linked SSBs in vivo, corroborating previous research and providing evidence that this is governed by primary DNA sequence at the cleavage site.

## Results

**CC-seq enriches Spo11-linked DNA fragments**. To elucidate the in vivo functions of topoisomerase-like enzymes, we set out to establish a method (termed 'CC-seq') to enrich and map protein-DNA CCs genome-wide with nucleotide resolution (Fig. 1a). Silica fibre-based enrichment of CCs was first developed to detect the adenovirus terminal protein[22] and was later used to isolate the meiotic DSB-inducer Spo11[23]. We first verified this enrichment principle using meiotic *sae2Δ* cells, in which Spo11-linked DSBs accumulate at defined loci due to abrogation of the nucleolytic pathway that releases Spo11[24]. Specifically, ethanol-fixed meiotic *S. cerevisiae* cells were lysed in the absence of proteolysis using strong protein-denaturing detergent at 65 °C and extracted with a phenol/chloroform mixture to remove noncovalently bound protein, generating an aqueous phase enriched in total genomic DNA and putative Spo11 CCs. At this stage, DNA fragmentation was avoided to minimise the generation, and loss, of any low molecular weight Spo11 CC DNA fragments that we anticipate will partition to the organic interphase[24]. Such purified genomic DNA was digested with *Pst*I restriction enzyme, and passed through glass-fibre spin columns in the presence of a high salt buffer that promotes protein binding[22,23] (and thereby Spo11 CC binding), but suppresses the binding of DNA not covalently linked to protein. Bound fractions were eluted with detergent and used to assay a known Spo11-DSB hotspot by Southern blotting (Fig. 1b). While DSB fragments were a minor fraction of input material (~10% of total), and were absent in wash fractions, DSBs accounted for >99% of eluted material, indicating ~1000-fold enrichment relative to non-protein-linked DNA (Fig. 1b).

**CC-seq reproducibly maps known Spo11-DSB hotspots genome-wide**. We next used our CC-enrichment method to generate a genome-wide map of Spo11-DSBs. Specifically, rather than fragmenting by restriction enzyme, equivalent non-proteolysed phenol-extracted genomic DNA from meiotic *sae2Δ* cells was sonicated to <400 bp in length, passed through the silica column, eluted, and ligated to DNA adapters in a two-step procedure that utilised the known phosphotyrosyl-unlinking activity of mammalian TDP2[25,26] to uncap the Spo11-bound end (Fig. 1a, Supplementary Fig. 1a). Use of TDP2 only after ligation of the first adapter generates: (1) specificity for 5′-linked CCs (Supplementary Fig. 1b); (2) nucleotide precision at the CC end; and (3) depletes the library of any contaminating free-DNA or DNA fragments with internally bound protein because such fragments will gain an identical adapter on both ends, preventing clustering on Illumina flow cells (Supplementary Fig. 1c). Notably, because sonication will break ssDNA opposite a nicked substrate[27], putative 5′-linked CC-SSBs will be converted into mappable DSB ends (Supplementary Fig. 1d). Thus, if present, CC-seq maps will be composite mixtures of 5′-linked CC-DSBs and 5′-linked CC-SSBs.

Libraries were paired-end sequenced and mapped to the *S. cerevisiae* reference genome (Supplementary Table 1) alongside reads from a previous mapping technique ('Spo11-oligo-seq') that isolates Spo11-linked oligonucleotides generated in wild-type cells during DSB repair[28]. CC-seq revealed sharp, localised peaks ('hotspots') in *SPO11*+ cells that visually (Fig. 1c) and

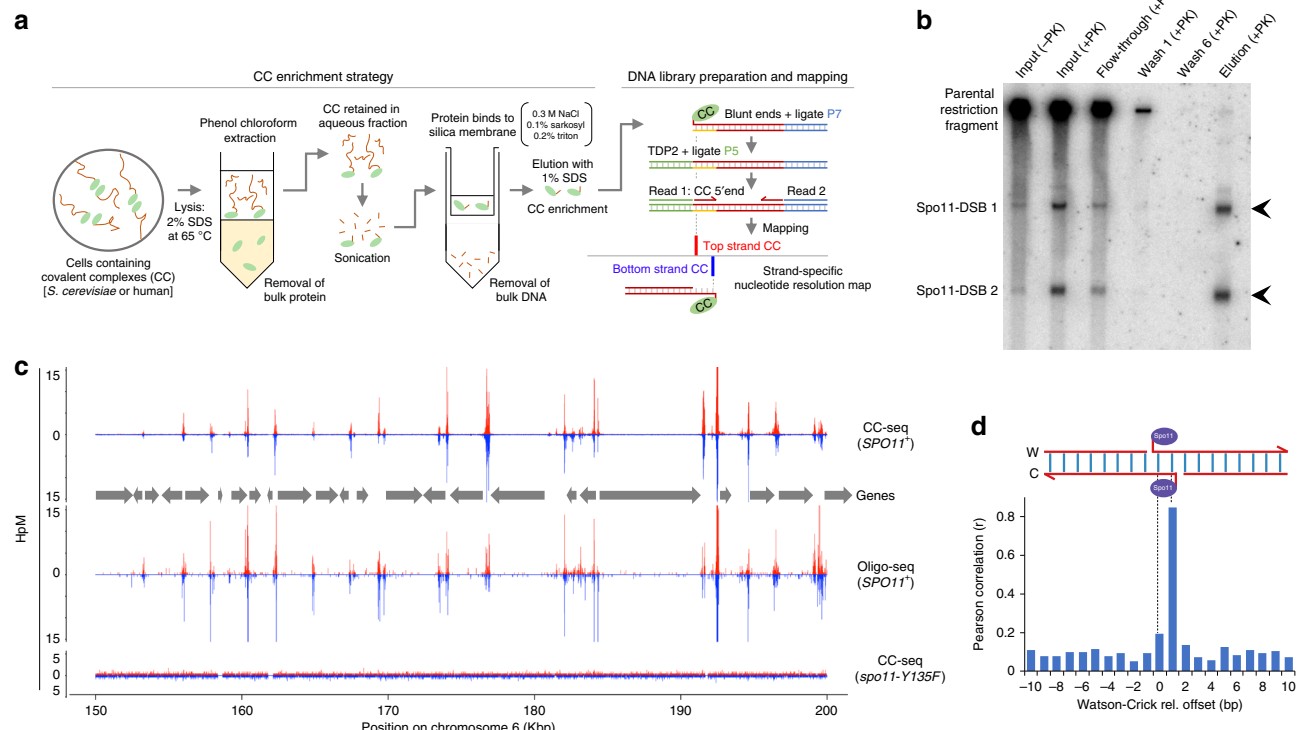

**Fig. 1** CC-seq maps covalent Spo11-linked DNA breaks in *S. cerevisiae* meiosis with nucleotide accuracy. **a** Schematic of the CC-seq method. **b** Silica-based enrichment of meiotic *S. cerevisiae* Spo11-linked DNA fragments detected by Southern blotting at the *his4::LEU2* recombination hotspot. Arrowheads indicate expected sizes of Spo11-DSBs. '+PK' = samples treated with proteinase K prior to electrophoresis. The eluate was loaded at 4X cell equivalents relative to the input. **c** Nucleotide resolution mapping of meiotic *S. cerevisiae* Spo11 hotspots by CC-seq or oligo-seq[28]. Red and blue traces indicate Spo11-linked 5′ DNA termini on the Watson and Crick strands, respectively. Grey arrows indicate positions of gene open reading frames. **d** Pearson correlation (*r*) of Spo11 CC-seq signal between Watson and Crick strands, offset by the indicated distances. Cartoon shows the known Spo11-DSB structure. HpM Hits per million mapped reads per base pair. Source data are provided as a Source Data file.

quantitatively (Supplementary Fig. 2a, *r* = 0.82) correlated with Spo11-oligo seq, and that were absent in the *spo11-Y135F* control strain in which Spo11-DSBs do not form[21], demonstrating that meiotic signal detected by CC-seq accurately reflects the distribution of Spo11 cleavages. Moreover, CC-seq biological replicates were highly correlated (Supplementary Fig. 2b; *r* = 0.98), demonstrating high reproducibility. Finally, when analysed at nucleotide resolution, CC-seq revealed high correlation (*r* = 0.85) between the frequency of Watson-mapping and Crick-mapping cleavage sites, when offset by a single bp (Fig. 1d and Supplementary Fig. 2c). This correlation is expected because of the 2 bp 5′ overhang generated at Spo11-DSBs[29] (Fig. 1d)—demonstrating the nucleotide resolution accuracy of CC-seq, which was further supported by nucleotide composition preferences at the site of cleavage (Supplementary Fig. 2d, e) similar to those reported previously[28].

**CC-seq maps Top2-DNA CCs in *S. cerevisiae*.** Confident in the specificity of CC-seq to map meiotic Spo11 activity with high resolution and dynamic range, we employed CC-seq to characterise within *S. cerevisiae* the CCs generated naturally by Topoisomerase 2 (Top2) that become stabilised upon exposure to VP16, and are thus a proxy for Top2 catalytic activity[30,31]. Because *S. cerevisiae* is relatively insensitive to VP16, we utilised strains ('*pdr1Δ*') in which the action of major drug export pathways are downregulated[32]. As expected, *pdr1Δ* dramatically increased cellular sensitivity to VP16, which was further enhanced by mutation of *SAE2* and *MRE11* (Fig. 2a), factors involved in the repair of covalent protein-linked DNA breaks[33–36].

Next, we prepared CC-seq libraries from vegetatively growing VP16-treated control, *sae2Δ*, and *mre11Δ S. cerevisiae pdr1Δ* cells (Supplementary Table 2). A human DNA spike-in was added to enable calibration of relative signal between *S. cerevisiae* CC-seq libraries ("Methods"). Replicate libraries displayed high reproducibility (*r* > 0.89; Supplementary Fig. 3a–c), so were pooled. In untreated control cells, weak CC-seq signal was distributed homogeneously across the genome, with few strong peaks (Fig. 2b). Upon VP16 treatment, sharp single-nucleotide peaks arose at similar locations in all strains (Supplementary Fig. 3d, e), with a visual tendency towards enrichment at intergenic regions (IGRs) (Fig. 2b). As expected, based on our genetic sensitivity assays (Fig. 2a), CC-seq signal was increased in MRX DNA repair pathway mutants (Fig. 2b and Supplementary Fig. 3e). Additionally, CC-seq signal was strongly correlated between Watson and Crick strands when offset by 3 bp (Fig. 2c and Supplementary Fig. 3f–k)—as expected for the 4 bp overhang generated at Top2-DSBs. Collectively these data describe how CC-seq generates a strand-specific nucleotide resolution map of VP16-induced Top2 CCs.

Consistent with visual inspection, global analyses revealed Top2 activity in *S. cerevisiae* to be relatively enriched in divergent and tandem IGRs (those IGRs containing gene promoters) and depleted in convergent IGRs and intragenic regions, similar to, but less pronounced than, the patterns of Spo11 (Fig. 2d, e). Such connections to global gene organisation are likely driven by the need for Top2 and Spo11 to interact with the DNA helix—an interpretation underpinned by an anticorrelation with nucleosome occupancy as measured by aggregating MNase-seq data[37] around both strong vegetative (Supplementary Fig. 4a) and

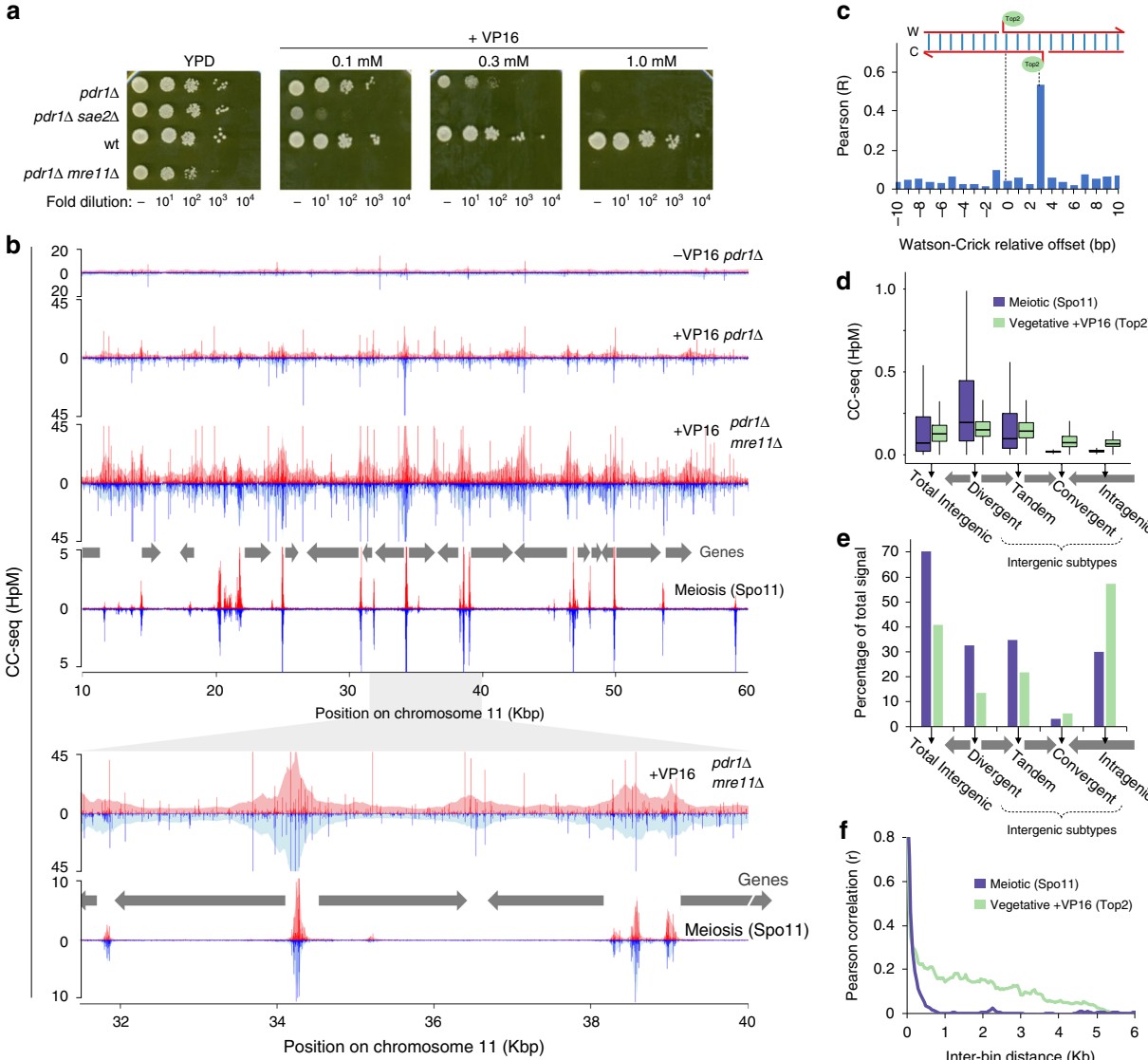

**Fig. 2** CC-seq maps covalent Top2-linked DNA breaks in vegetative *S. cerevisiae* with nucleotide accuracy. **a** Serial dilution spot tests of VP16 tolerance for the indicated strains. **b** Nucleotide resolution CC-seq maps of vegetative *S. cerevisiae* Top2 CCs in the indicated strains after 4 h treatment with 1 mM VP16. Spo11 CC-seq data is plotted for comparison. Top2 CC-seq data were calibrated using a human DNA spike-in. Red and blue traces indicate CC-linked 5′ DNA termini on the Watson and Crick strands, respectively. Grey arrows indicate positions of gene open reading frames. Lower panels show an expanded view of the region from 31.5 to 40 Kbp. Pale shaded areas are the same data smoothed according to local density. **c** Pearson correlation (*r*) of Top2 (*pdr1Δ*) CC-seq signal on Watson and Crick strands, offset by the indicated distance. **d** Quantification of vegetative Top2 (*pdr1Δ*) and meiotic Spo11 CC-seq signal, stratified by genomic region. The genome was divided into intra and intergenic regions; the latter further subdivided into divergent, tandem and convergent based on orientation of flanking genes. Spo11 and Top2 activity mapped by CC-seq is expressed as box-and-whisker plots of density (upper and lower box limit: third and first quartile; bar: median; upper and lower whisker: highest and lowest values within 1.5-fold of the interquartile range). Wilcox *p*-values < $10^{-3}$. **e** Aggregation of data shown in (**c**), expressed as the percentage of total mapped reads. **f** Local autocorrelation of Top2 (*pdr1Δ*) or Spo11 CC-seq signals. Top2 or Spo11 CC-seq data were binned at 50 bp resolution and the Pearson correlation (*r*) calculated between bins of increasing separation. HpM Hits per million mapped reads per base pair

meiotic (Supplementary Fig. 4b) CC-seq peaks, similar to prior observations[28,38]. Nevertheless, Top2 CC and Spo11 CC patterns are far from synonymous. Spo11 activity is focused almost entirely at promoter-containing IGRs, whereas Top2 CC signal was also detected at convergent IGRs, and within intragenic regions (Fig. 2d, e). Moreover, Top2 CC signal remained weakly autocorrelated over greater distances than Spo11 (Fig. 2f), suggesting localised 5–10 kb wide regions of enriched activity—much larger than Spo11-DSB hotspots[28] (generally <500 bp in size; Fig. 2b)—suggesting factors other than just nucleosome occupancy influence local Top2 catalysis in *S. cerevisiae*.

Aggregating Top2 CC-seq data around TSSs revealed a narrow peak centred ~100 bp upstream of the TSS (Supplementary Fig. 4c), with weaker periodic signal also present within the gene body. Overlaying MNase-seq data[37] revealed these patterns to directly correspond to the narrow NDR present upstream of annotated *S. cerevisiae* TSSs[39], and an anticorrelation with the well-positioned nucleosome ordering present within genes (Supplementary Fig. 4c). By contrast, and in agreement with previous studies[28], Spo11 signal was excluded from genic sequences (Supplementary Fig. 4d), instead found exclusively within, and matching the average profile of the NDR

(Supplementary Fig. 4d), which was wider in meiotic cells. Importantly, despite the preferential association of Top2 CC around TSSs (Supplementary Fig. 5a), stratification by mRNA expression[40] revealed no quantitative correlation—neither following VP16 exposure (Supplementary Fig. 5b–d), nor in untreated cells (Supplementary Fig. 5e, f). Equivalent analysis of CC-seq in meiotic cells similarly revealed no quantitative correlation between Spo11 activity and proximal gene expression (Supplementary Fig. 5g–j), as previously concluded[41].

**CC-seq maps TOP2-DNA CCs in human cells**. Having demonstrated that CC-seq is applicable for mapping two distinct types of topoisomerase-like CCs in *S. cerevisiae*, we applied CC-seq to map TOP2 activity in a human cell system. Asynchronous, sub-confluent RPE-1 cells (non-cancer, immortalised, diploid karyotype) were treated with VP16 in the presence of MG132 proteasome inhibitor, in order to limit potential proteolytic degradation of TOP2 CCs that might otherwise hamper enrichment. Slot-blotting and immunodetection with anti-TOP2β antibody demonstrated complete recovery of input TOP2β-linked CCs within the column elution, with no TOP2β remaining in the flow through or being removed by the washes (Fig. 3a). Eluted material was used to generate sequencing libraries from three replicate control (−VP16) and four replicate VP16-treated (+VP16) samples, and high-depth paired-end sequencing reads (Supplementary Table 2) were aligned to the human genome (hg19).

Replicates displayed high correlation in the distribution of TOP2 CC signal at broad scale (r values ≥ 0.79; Supplementary Fig. 6a, b), and so were pooled. Visual inspection revealed that −VP16 and +VP16 signals were spatially-correlated (Supplementary Fig. 6a), but that +VP16 signal intensity was less uniform, with more dynamic range (Supplementary Fig. 6a). These visual trends were confirmed by global comparisons of 10 kb-binned data ± VP16 (Supplementary Fig. 6c, d). Such differences are consistent with the small amount of TOP2β signal detected by slot blotting in untreated cells (Fig. 3a), which leads to a higher signal-to-noise ratio following VP16 treatment. Like yeast Top2 libraries, nucleotide resolution analysis of human CC-seq libraries displayed a skew towards Watson-Crick read-pairs that are offset by 3 bp, as expected for the 4 bp 5′ overhang generated by TOP2 (Supplementary Fig. 6e).

To demonstrate the ability of CC-seq to map both human TOP2α and TOP2β activities we used CRISPR-Cas9 to generate a human RPE-1-derived cell line with homozygous knockout mutations in the non-essential *TOP2B* gene, and arrested cells in G1 (Supplementary Fig. 7a) when TOP2α is not expressed[42]. TOP2α and TOP2β expression was undetectable in such G1-arrested cells (Supplementary Fig. 7b), and importantly, immunostaining revealed VP16-induced γ-H2AX foci (markers of DSBs) were reduced ~7-fold relative to wild type control lines, indicating that most of the DSB formation is TOP2β-dependent (Supplementary Fig. 7c, d). Next, we applied CC-seq to asynchronous and G1-arrested wild type and *TOP2B*$^{-/-}$ cells. In G1, VP16 exposure induced localised regions of enriched TOP2 CC formation in wild type cells that were largely diminished or absent in *TOP2B*$^{-/-}$ cells, whereas TOP2β deletion did not prevent VP16-induced signal in asynchronous cells, in which TOP2α is still present (Supplementary Fig. 7e). Together, these results indicate that CC-seq detects a mixture of both TOP2α and TOP2β CCs in wild type human cells depending on the cell cycle phase.

As a final proof of principle, we wished to compare CC-seq sites with genomic maps of VP16-induced DNA breaks in human cells recently generated using a different methodology,

‘END-seq’[18], that is not specific for protein-linked TOP2 CC. Aggregating END-seq data around strong CC-seq positions revealed concordant peak signals at broad scale (Fig. 3b), demonstrating a general agreement in the positions reported by the two techniques. However, at finer scale, END-seq positions were offset by 1–15 bp in the 3′ direction relative to CC-seq positions on each strand (Fig. 3c)—something not observed when aggregating CC-seq data around the same loci (Fig. 3c). This important difference is likely to be explained by the different populations of DNA breaks mapped by each technique: specific TOP2 cleavage positions (CC-seq) versus DNA breaks that have undergone limited nucleolytic resection, either in vivo or during library preparation (END-seq).

**Spatial distribution of TOP2 activity in the human genome**. The human genome is divided into a relatively gene-rich 'A' compartment, and a relatively gene-poor 'B' compartment[43]. In both untreated and VP16-treated asynchronous wild type cells, Top2 activity was higher in the active A compartment (Fig. 3d, e), and also aligned with regions of negative DNA supercoiling[44] (Fig. 3d). Closer visual inspection (Fig. 3f) suggested that TOP2 CC-seq signals were enriched around genome features previously identified as regions of high TOP2-binding and VP16-induced DNA strand breakage, including binding sites of the genome-organising factor CTCF[45] and active TSSs marked by H3K4Me3 and H3K27Ac[20,46].

**CTCF-proximal TOP2 activity in the human genome**. To investigate the relationship between CTCF and TOP2 activity, we filtered CTCF consensus sequence binding motifs to include only those that overlapped an RPE-1 CTCF ChIP-seq peak[12]. After aligning Watson and Crick motifs in the same orientation, we frequently observed a strong TOP2 CC peak upstream of the motif centre accompanied by more distal, weaker peaks on both sides of the motif (Fig. 4a and Supplementary Fig. 8a). Loci containing multiple CTCF-binding motifs display more complex TOP2 CC patterns (Fig. 4a and Supplementary Fig. 8b) with enrichment on both sides of the double motif, as would be expected from the aggregation of a heterogeneous population of CTCF and TOP2 activity present across different cells.

We next aggregated CC-seq signals centred on oriented CTCF motifs that were stratified into quantiles based on proximal CTCF binding intensity (Fig. 4b, c). Most Top2 CC signal was concentrated in two peaks located ±~54 bp relative to the centre of the CTCF motif, with the upstream peak ~2-fold more intense (Fig. 4b, c). Total aggregated TOP2 CC signal across these regions is positively correlated with CTCF occupancy as measured by ChIP-seq (Fig. 4b–d). Additional Top2 CC signal is concentrated in a flanking array of weaker peaks that anticorrelate with the well-positioned nucleosomes at these sites (Fig. 4e). A similar, but much weaker, pattern of CC-seq was detectable in untreated cells, consistent with our prior interpretation that CC-seq enriches for a low frequency of spontaneous TOP2 CCs (Fig. 4e). Notably, anticorrelation with nucleosome positioning is not a unique feature of CTCF loci, but is instead a general property of TOP2 CC sites (Fig. 4f), consistent with maps of both Top2 CC (Supplementary Fig. 4a) and Spo11[28] (Supplementary Fig. 4b) in *S. cerevisiae*.

The CTCF-proximal Top2 CC pattern is similar to the pattern reported recently for VP16-induced DSBs[18], and has been suggested to result from a role for TOP2 in facilitating genome organisation by cohesin-dependent loop extrusion operating in the context of CTCF and other boundary elements. Interestingly, however, aggregated TOP2 CC-seq signal centred on loop-anchor

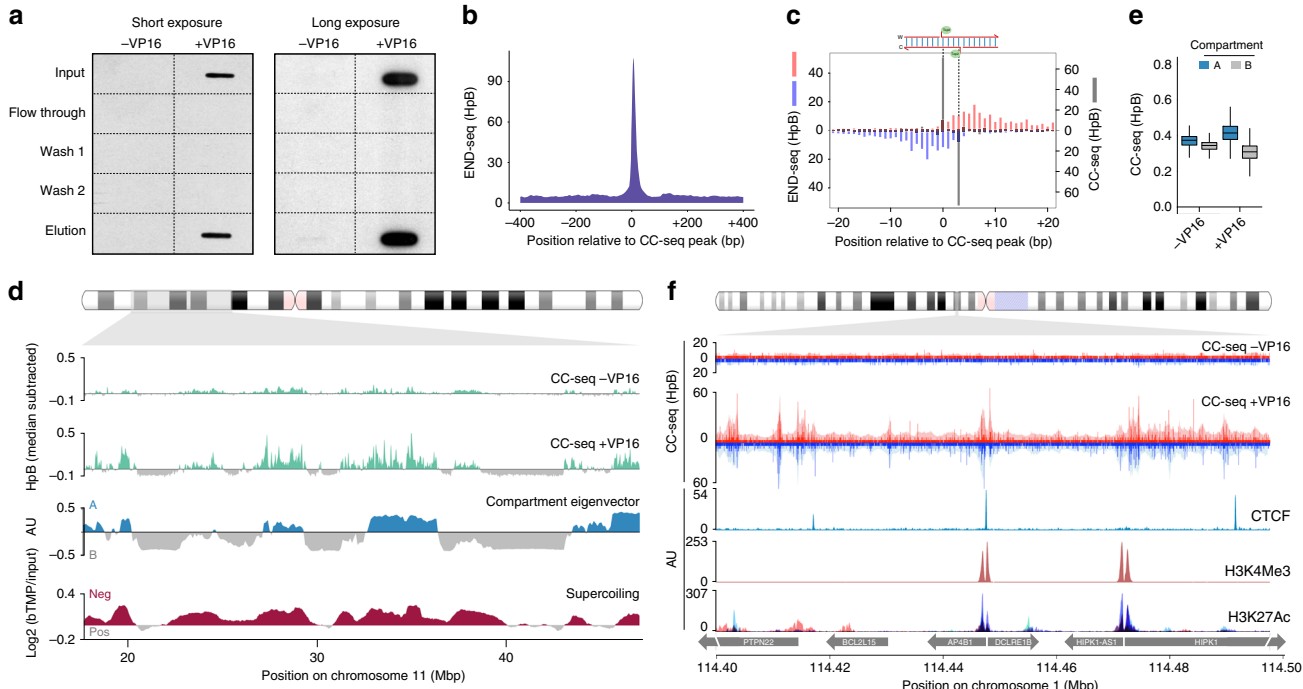

**Fig. 3** CC-seq maps TOP2-linked DNA breaks in human cells with nucleotide accuracy. **a** Anti-TOP2β western slot blot of input, flow through, wash, and elution fractions. RPE-1 cells were treated or not with 100 μM VP16, prior to processing according to Fig. 1a. **b** Medium-scale aggregate of END-seq mapped DSBs[18] surrounding nucleotide resolution CC-seq mapped TOP2 CCs. **c** Fine-scale strand-specific aggregate of END-seq mapped DSB ends (red and blue bars) and nucleotide resolution CC-seq mapped TOP2 CCs (grey bars) surrounding strong TOP2 CC sites. **d** Broad-scale CC-seq maps of TOP2 CCs produced in human RPE-1 cells ± VP16. Raw data were scaled, binned, smoothed and median subtracted prior to plotting. Chromatin compartments revealed by Hi-C eigenvector analysis[47], and supercoiling revealed by bTMP ChIP-seq[44] are shown for comparison. **e** Quantification of TOP2 CCs in chromatin compartments A and B. Data are expressed as box-and-whisker plots of density as for Fig. 2d. **f** Fine-scale CC-seq maps of TOP2 CCs produced in human RPE-1 cells ± VP16. Red and blue traces indicate TOP2-linked 5′ DNA termini on the Watson and Crick strands, respectively. Data in pale shaded areas has been smoothed according to local density. RPE-1 CTCF and H3K4Me3 ChIP-seq data plus H3K27Ac ChIP-seq data overlaid from seven cell lines[71] is shown for comparison. HpM, HpB Hits per million (or billion) mapped reads per base pair. Source data are provided as a Source Data file

CTCFs annotated previously by Hi-C[47] were not correlated with interaction strength of associated loops (Supplementary Fig. 8c).

**TSS-proximal TOP2 activity in the human genome**. Next, we focussed on the VP16-induced TOP2 CCs enriched around human TSSs. When assessing individual loci, we frequently observed CC-seq signal concentrated in broad regions both upstream and downstream of the TSS (Fig. 5a). We next aggregated CC-seq signals centred on TSSs stratified into quantiles based on gene expression microarray data[48] (Fig. 5b–d). On average, CC-seq signal was located in two broad peaks: a weaker peak located ~0.75 kb upstream and a stronger peak located ~1.0 kb downstream of the TSS (Fig. 5b–d), similar to prior reports of VP16-induced DSBs in mouse cells[20]. These signals were stronger and spread further from the TSS in more highly expressed genes (Fig. 5b, c). Quantification of CC-seq signal revealed quantitative correlations with gene expression as measured by microarray (Fig. 5e), transcription initiation as measured by CAGE[49] (Fig. 5f), and nascent transcription as measured by GRO-seq[50] (Fig. 5g, h). Increased Top2 activity was also positively associated with increased gene length, independent from gene expression (Supplementary Fig. 9a–c). Zooming in to the TSS-proximal region revealed a ~100 bp wide zone of depleted CC-seq signal ~50 bp upstream of the TSS, flanked on the right by a sharp peak of Top2 CC-seq signal centred ~30 bp downstream of the TSS that correlated with gene expression (Fig. 5d). Further downstream, within the transcribed region, a repeating array of Top2 CC-seq peaks are present separated by ~200 bp (Fig. 5d).

To investigate how these patterns correlate with nascent transcription and nucleosome structure, we overlaid aggregated maps of CC-seq, GRO-seq and MNase-seq for the top quartile of expressed genes (Fig. 5i). At broad scale, nucleosome depletion increased towards the TSS (Fig. 5i). In regions at least 1 kb from the TSS, CC-seq signals were positively correlated with this increasingly strong zone of nucleosome-depletion (Fig. 5i). However, within ~1.75 kb region demarcated by the summit of the two strong CC-seq regions, this relationship was lost, with relatively little CC-seq signal across the region most depleted for nucleosomes (Fig. 5i). GRO-seq, which quantitatively maps nascent transcription by RNA Pol II (RNAPII), is focussed in two narrow peaks of differential magnitude within this broad region of nucleosome depletion (Fig. 5i), reflective of sense and antisense transcription[51]. The relative magnitude of these GRO-seq peaks and the pattern of associated downstream signals matched the relative strength of the two broad flanking regions of CC-seq signal (Fig. 5i). Collectively, these features are suggestive of a quantitative and spatial relationship between transcription and Top2 activity.

We next investigated how these patterns are related at finer scale—directly at the site of transcription initiation (Fig. 5j). The TSS is located within a ~250 bp region of greatest nucleosome depletion, referred to here as the core NDR, which encompasses the known site of RNAPII loading and a region of limited nascent transcription[51] identified by GRO-seq (Fig. 5j). GRO-seq signal peaks ~60 bp downstream of the TSS (Fig. 5j)—indicative of promoter-proximal pausing (PPP) prior to the +1 nucleosome[51]. Interestingly, whilst Top2 CC-seq signal was depleted within the

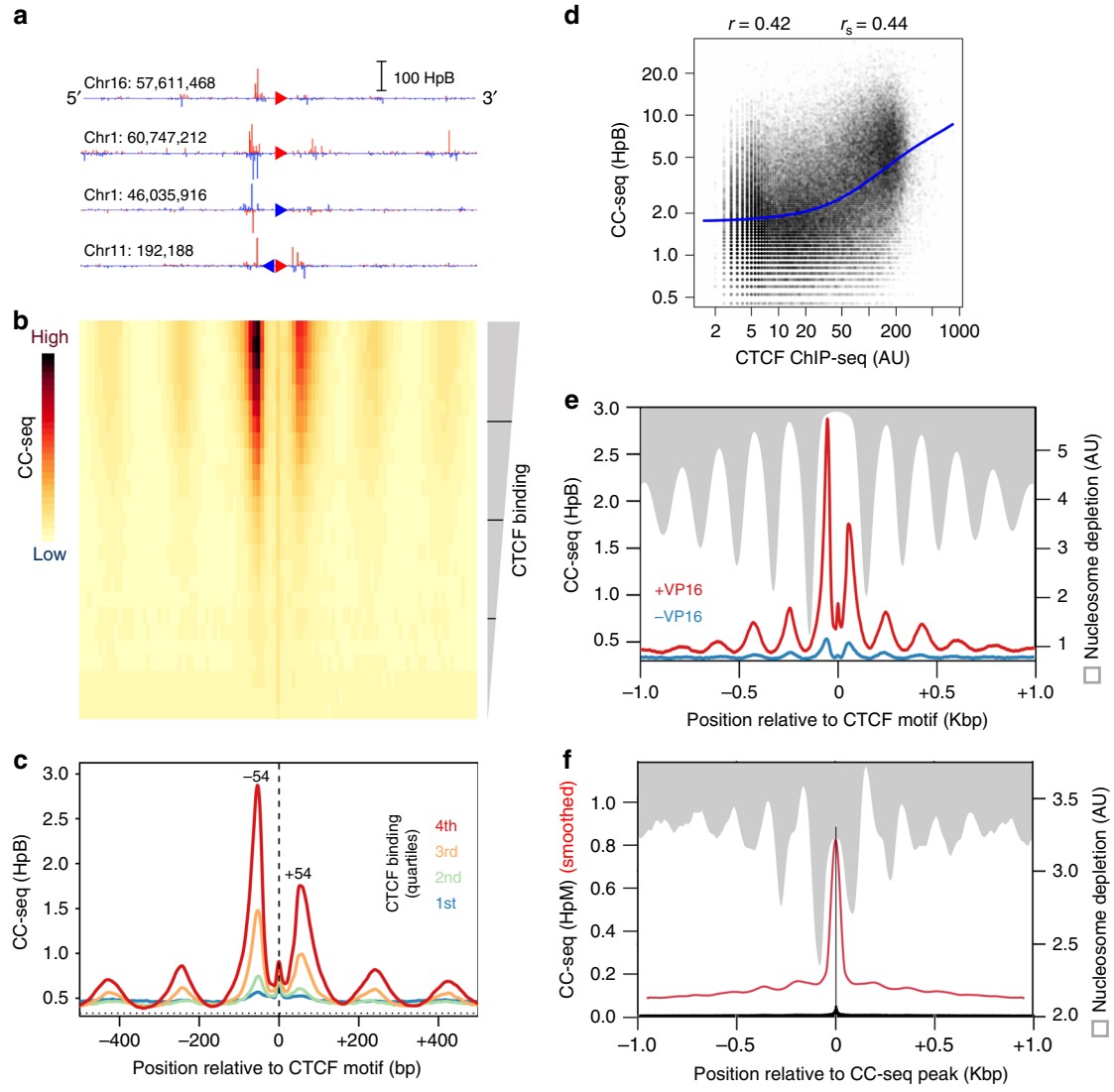

**Fig. 4** CTCF-proximal TOP2 activity is tightly confined within nucleosome-depleted regions. **a–c** Aggregation of TOP2 CCs in a 1 Kbp window centred on orientated CTCF motifs in human RPE-1 cells. Four example CTCF loci are shown orientated in the 5′-3′ direction. Red and blue triangles indicate Watson and Crick CTCF motifs, respectively (**a**). A heatmap of all CTCF-motifs in the human genome, with 25 aggregated rows stratified by the strength of colocalising CTCF ChIP-seq peaks in RPE-1 cells. The colour scale indicates average TOP2 CC density in each 10 bp bin (**b**). Motifs are also stratified into quartiles of CTCF-binding, and the average TOP2 CC distribution in each quartile plotted. Horizontal line is the genome mean. Vertical line is the TSS (**c**). **d** Quantitative relationship between CC-seq (±100 bp) and CTCF ChIP-seq signal centred on CTCF motifs. Blue line = lowess curve. Pearson $r$ and Spearman $r_s$ values were calculated between CC-seq and log-transformed CTCF ChIP-seq. **e** Aggregated TOP2 CC distribution (red line) in the highest quartile of CTCF-binding compared with the average MNase-seq signal[71] (grey). **f** Aggregated TOP2 CC distribution (black and red lines: single-nucleotide resolution and smoothed, respectively) and the average MNase-seq signal (grey) surrounding strong TOP2 CC sites. MNase-seq data in (**e–f**) is inverted to emphasise spatial relationship between CC-seq and nucleosome depletion (white). HpM, HpB Hits per million (or billion) mapped reads per base pair

core promoter (−50 bp), the sharp central peak of Top2 CC-seq signal identified earlier (+30 bp) directly overlaps the region of limited nascent transcription immediately downstream (0–60 bp).

**CC-seq detects Top2 CC sites that display strand disparity.** Previous reports suggest that VP16 induces many SSBs, as well as DSBs, due to independent poisoning of each active site in the Top2 homodimer[17]. In both human and yeast, there is an enrichment of Top2 CC-seq signals on the Watson and Crick strands that are 3 bp offset (Fig. 2c, Supplementary Fig. 6e, Supplementary Fig. 10a and Supplementary Fig. 11a). Interestingly, however, visual inspection also revealed many genomic sites with obvious disparity: where 3 bp offset Watson and Crick Top2 CC frequencies were imbalanced (Supplementary Fig. 10a

and Supplementary Fig. 11a). Because preferential sonication opposite nicks[27] will convert a Top2 CC-SSB into a DSB end that will be included in the CC-seq library (Supplementary Fig. 1d), we reasoned that sites with strong disparity might indicate locations of strand-biased SSB formation.

To investigate this possibility, we compared observed CC-seq data to a simulated model where all sites have Watson and Crick parity (termed 'cognate' sites) and are sampled at random (Fig. 6a and Supplementary Fig. 10b). In this model, we hypothesise that variable visibility in the experimental data arises due to independent sampling of Watson and Crick reads from the population, rather than from a real biological difference. Notably, both human and yeast datasets had many more sites with significant disparity (two-sample, two-sided Poisson test, $p < 0.05$; 'non-cognate sites') than expected by our simulated 'cognate'

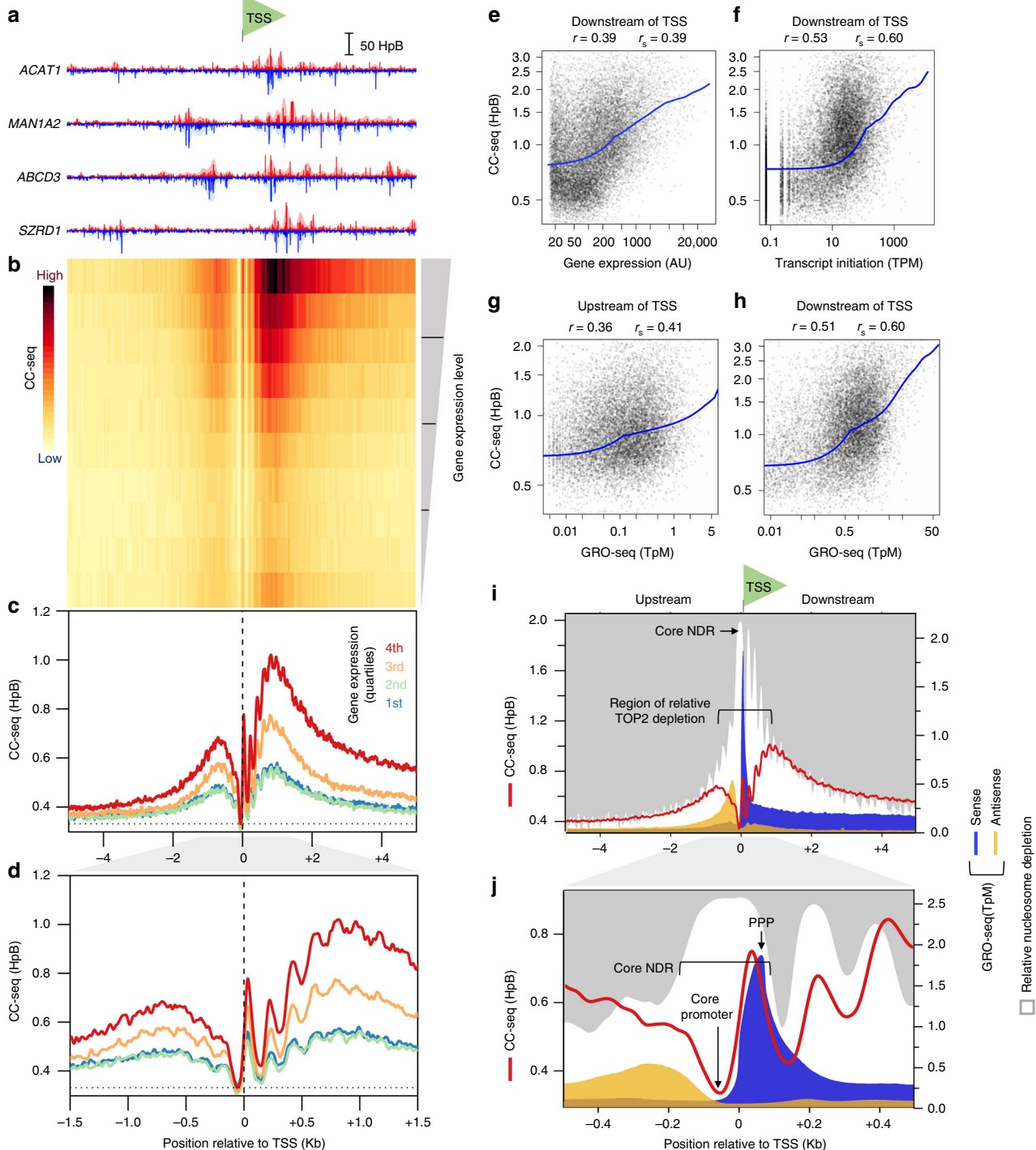

**Fig. 5** TSS-proximal TOP2 activity is strongly correlated with gene transcription. **a–d** Aggregation of TOP2 CCs in a 10 Kbp window centred on orientated TSSs in human RPE-1 cells. Four example TSSs are shown orientated in the 5'–3' direction (**a**). A heatmap of all TSSs in the human genome, with 10 aggregated rows stratified by RPE-1 gene expression level[48]. The colour scale indicates average TOP2 CC density in each 50 bp bin (**b**). Motifs are also stratified into 4 quartiles of gene expression, and the average TOP2 CC distribution in each quartile plotted at two levels of zoom. Horizontal line is the genome mean. Vertical line is the TSS (**c–d**). **e–h** Quantitative correlation between CC-seq signal and transcription rate. Scatterplot of CC-seq (5 Kbp window downstream of TSS) and gene expression (**e**), or transcription initiation[49] (**f**). Scatterplot of CC-seq signal in 5 kb windows directly upstream and downstream of TSS and GRO-seq in 200 bp regions directly upstream and downstream of TSS. Blue line = lowess curve. Pearson r and Spearman $r_s$ values were calculated between CC-seq and log-transformed measures of transcription rate. **i–j** Aggregate comparison of CC-seq (red line), nucleosome occupancy[71] (grey shading), and strand-specific GRO-seq[50] centred on TSS at two levels of zoom. MNase-seq data is inverted to emphasise spatial relationship between CC-seq and nucleosome depletion (white). PPP promoter proximal transcriptional pausing, TpM Transcripts per million mapped reads per base pair, HpB Hits per billion mapped reads per base pair, AU arbitrary units

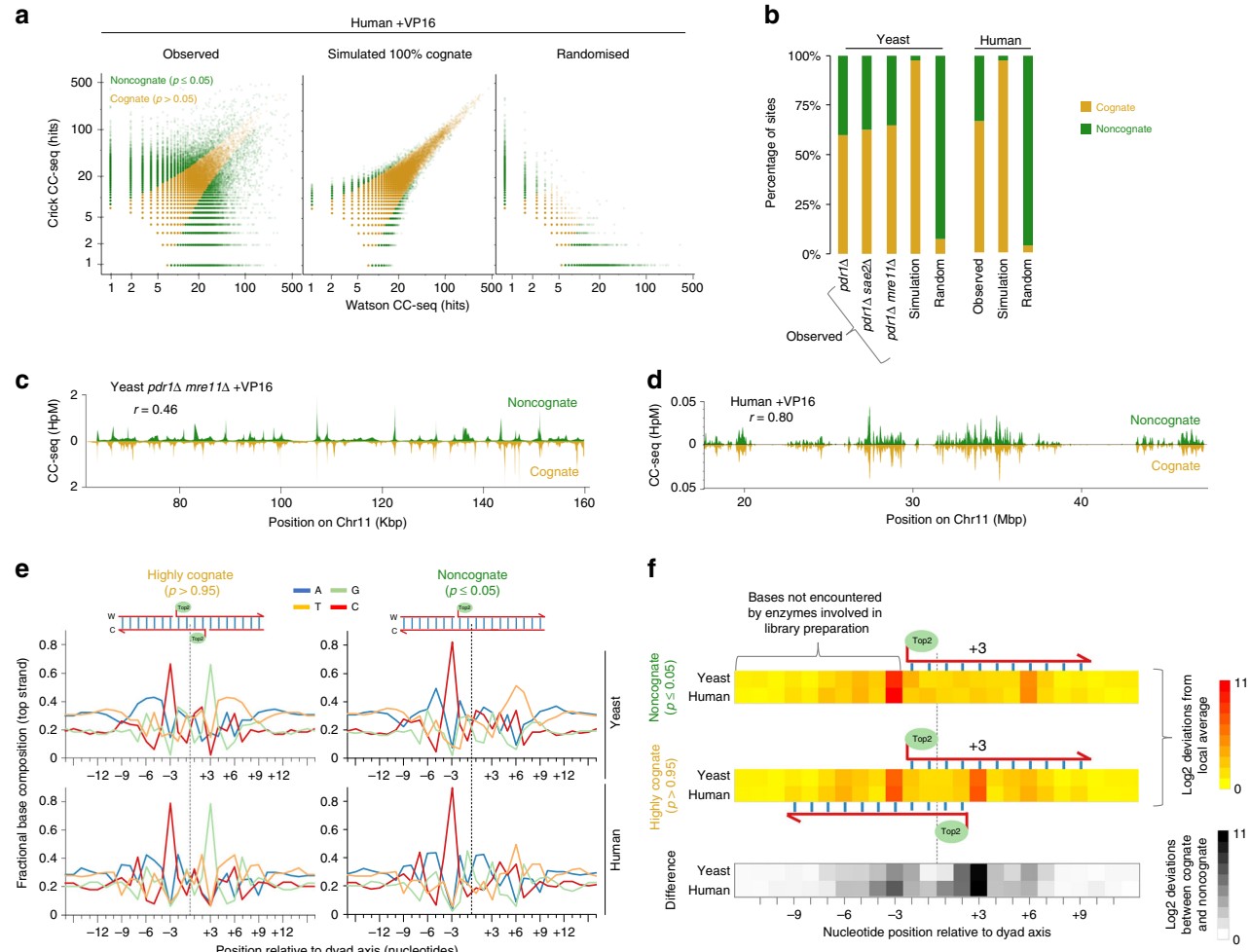

**Fig. 6** Local DNA sequence composition deviations associated with cognate and noncognate CC-seq sites. **a** Scatterplots of human TOP2 CC-seq hits on the Watson strand vs the Crick strand at each 3 bp offset site, in: observed data; a dataset sampled from a simulation where all 3 bp offset sites had identical Watson and Crick values ('100% cognate'); randomised observed data where Crick hits were shuffled. Noncognates (green) and cognates (gold) were defined as those 3 bp offset sites with significantly different, or not significantly different Watson and Crick hits, respectively (two-sample, two-tailed Poisson test; $p \leq 0.05$). Sites with fewer than eight hits in total (Watson + Crick) were excluded due to insufficient statistical power. **b** Percentage of Top2 CC-seq sites that were classed cognate vs noncognate in: observed, simulated 100% cognate, and randomised as defined in (**a**) for *S. cerevisiae* and human VP16-treated samples. *S. cerevisiae* cognate simulation and randomisation use *pdr1Δ mre11Δ* dataset. **c, d** Cognate (gold) and noncognate (green) Top2 CC-seq signal from VP16-treated *pdr1Δ S. cerevisiae* cells (**c**) or RPE-1 cells (**d**), within example regions. The data were smoothed according to local density (**c**), or binned at 10 Kbp resolution and smoothed with a sliding 10-bin Hanning window (**d**). **e** Average nucleotide composition over a 32 bp window centred on the highly cognate ($p > 0.95$) or (inferred) noncognate ($p \leq 0.05$) Top2 dyad axis, in VP16-treated *pdr1Δ mre11Δ S. cerevisiae* and human RPE-1 cells. Values reported are for the top strand only. **f** Heatmaps of nucleotide composition deviations from the local average, for noncognate (top) and highly cognate (middle) sites; and the difference between the two (bottom). For the upper two panels, the absolute $\log_2$ deviations of the four nucleotides were summed at each position, and are expressed on a colour-scale from yellow to red. For the bottom panel, the absolute $\log_2$ fold difference between highly cognate and noncognate were summed at each position and are expressed on a colour-scale from white to black. HpM Hits per million mapped reads per base pair

model (Fig. 6a, b and Supplementary Fig. 10b). At the same time, cognate sites were also still far more frequent in real data than were present in a randomised dataset in which the frequency of Watson and Crick reads within 3 bp offset sites were shuffled relative to one another (Fig. 6a, b and Supplementary Fig. 10b). From these analyses we infer that while Top2 CCs do preferentially arise as cognate pairs, there are some genomic sites where Top2 CC are enriched on one strand ('noncognates') —potentially consistent with strand-biased SSB formation.

Despite inherent fine-scale differences in their genomic locations, broad-scale patterns of cognates and noncognates were correlated in both yeast and human datasets, suggesting that both are equivalent measures of Top2 activity (Fig. 6c, d, Supplementary Fig. 10c and Supplementary Fig. 11b). Moreover, as when considering all genomic CCs, *S. cerevisiae* noncognate sites

remained preferentially associated with intergenic DNA (Supplementary Fig. 10d) and were enriched upstream of TSSs (Supplementary Fig. 10e), whilst human noncognate sites remained preferentially associated with compartment A (Supplementary Fig. 11c) and positively correlated with transcription (Supplementary Fig. 11d).

**Local DNA sequence directs the formation of Top2 CCs.** Prior, fine-scale analysis of Top2 cleavage within specific substrates indicated that cleavage patterns are influenced by local DNA sequence composition[52,53]. To investigate the generality of such findings, and to precisely determine how DNA sequence influences Top2 site preference across the yeast and human genome in vivo, we utilised the nucleotide resolution accuracy of CC-seq

to compute the average base composition around Top2 CC sites (Supplementary Fig. 12a–c). In both human and *S. cerevisiae* datasets we observed base composition preferences around the Top2 cleavage site, revealing a core region (±10 bp) of stronger bias ($p < 10^{14}$, Chi-squared test), consistent with the region of DNA occluded by Top2[54–56], plus flanking regions (±10–40 bp) of weak ~10.5 bp periodic bias (Supplementary Fig. 12b, c), most evident in the A and T bases (Supplementary Fig. 12d, e).

We next fractionated Top2 CC-seq data into cognate and noncognate sites and focused on the core region of stronger sequence bias (Fig. 6e). For highly cognate locations ($p > 0.95$), average DNA sequence patterns are rotationally symmetrical around the dyad axis of cleavage (Fig. 6e), consistent with the known homodimeric nature of eukaryotic Top2[1]. At positions −3 and +3 on the top strand there is a strong preference for cytosine and guanosine, respectively (Fig. 6e). This equates to a preference for cytosine at the bases immediately 5′ to the scissile phosphodiester bonds cleaved by Top2. Importantly, whilst noncognate sites display similar overall symmetrical sequence features as cognates (Fig. 6e), noncognates differed substantially at position +3, where the strong preference for guanosine on the top strand was absent (Fig. 6f), which is consistent with a 5′ cytosine being strongly associated with cleavage. Critically, the 5′ cytosine skew at the −3 position is strong at both cognate and noncognate sites. This, and other upstream bases, are never encountered by the enzymes used to generate CC-seq libraries and therefore must be a biological indicator of preferred Top2 cleavage. Thus, overall, the unique sequence composition at noncognate positions leads us to favour the view that they indicate preferred sites of VP16-induced SSBs in vivo.

## Discussion

Topoisomerases facilitate changes to DNA topology which are essential for genome maintenance and replication. Yet, topoisomerase dysfunction can also prove harmful, generating protein-linked DNA breaks that create risks to genome stability. Thus, understanding how, where, and when topoisomerase activity takes place is of great interest. A range of methods have been developed for mapping DSB ends[18,19,57–59], though none specifically enrich for protein-DNA CCs. Moreover, many of these techniques do not achieve nucleotide resolution accuracy for Top2 CC positions, either due to the use of nucleolytic processing to blunt 5′ DNA termini, and/or inability to directly map DNA breaks covalently linked to protein[18,19,57,58]. Here, by first enriching CCs and then utilising the tyrosyl phosphodiesterase activity of recombinant TDP2 to unblock 5′-DNA termini non-nucleolytically, we have developed a sensitive method to map sites of preferred type IIA topoisomerase activity across the eukaryotic genomes of yeast and human cells with strand-specific nucleotide accuracy.

Top2 activity has long been associated with transcriptional activity[8,20,46,60]. Yet, one of our most intriguing findings is that whilst Top2 is spatially patterned by gene location in both yeast and human cells, a local correlation with transcription rate was only present in the latter. The reason for this evolutionary difference is unclear but may stem from the very different scale and organisation of the yeast and human genomes. Our results would argue that topological stress—arising from transcription and/or other DNA metabolic processes—is largely resolved by Top2 in *S. cerevisiae* at intergenic regions (IGRs) regardless of where stress may be first generated. Because IGRs in yeast are frequent and regularly spaced regions of nucleosome depletion, we hypothesise that their preferential accessibility may mean most of the strand passage activity driven by Top2 arises where IGR sites cross (Fig. 7).

By contrast, the connection between Top2 activity and transcription in human cells was quantitative and distributed rather differently—focussed within gene bodies directly downstream of the TSS (Fig. 7). Due to the local correlation of Top2 activity with gene length and with transcription as measured by three independent assays, we favour the view that such CC-seq patterns are read-outs of the local topological stresses generated by transcriptional processes. Importantly, however, superhelical stress can be resolved by both Top2 and Top1. Thus, an intriguing possibility is that in *S. cerevisiae*, Top1 activity, which we do not map here, may be responsible for locally resolving transcriptional topological stress—a role that may be shared more equally by TOP2 in human cells.

Based on the twin-domain model of DNA supercoiling, topological stress is expected to increase not just with gene expression level, but also with increasing gene length[61]. In support of this idea, we observed a positive association between human TOP2 CC enrichment and gene length. Interestingly, no such correlation with gene length is present in *S. cerevisiae* (Supplementary Fig. 9d–f), further supporting a model whereby, in this compact gene-dense genome, Top2 activity in the IGR is not directly linked to expression nor length of the proximal gene.

As described by others[18,45], we find that TOP2 is also preferentially active in close proximity to the genome organising factor, CTCF (Fig. 7)—a relationship that we determined is quantitatively correlated with CTCF binding. Whilst it has been tempting to postulate that this association reveals a role for TOP2 in the process of loop extrusion[18], our inability to detect any quantitative correlation between loop boundary strength and CC-seq strength suggests that the relationship may be less direct. Given the requirement of Top2 to interact with the DNA helix in both yeast and human cells, perhaps the distinct patterning of nucleosomes generated by CTCF binding simply generates preferential opportunities for TOP2 to gain access to the DNA—similar to the situation in yeast IGRs for both Top2 and Spo11—with the processes generating torsional stress located elsewhere.

The reason why a prior study (END-seq[18]) failed to detect a connection between VP16-induced DSB formation and transcription is unclear. Our reanalysis of these published data indeed identifies only a weak trend (Supplementary Fig. 13a–c), with CTCF-associated signal relatively more prominent than transcription-associated signal than it is in our CC-seq libraries (Supplementary Fig. 13d–e). Such differences may indicate real biological differences between the cell lines analysed (MCF7 vs RPE-1 cells), or more intriguingly, differential technical sensitivity of END-seq and CC-seq to CTCF-proximal versus TSS-proximal TOP2 activity. For example, CCs generated at TSS-proximal regions may be more stable than at CTCF sites, increasing their visibility as measured by CC-seq.

The high spatial resolution of CC-seq also permits us to describe fine-scale features of Top2 activity—revealing that nucleosome occupancy is not the only determinant. Specifically, whilst human TSSs are centred within a relatively broad region of nucleosome depletion, TOP2 activity actually decreased towards its centre. Remarkably, however, within the core NDR, Top2 activity was focussed directly over the region of nascent transcription associated with promoter proximal pausing. We favour the view that despite topological stress most likely being present right across the TSS region, local occlusion by factors such as the RNAPII complex bound at promoters modulates TOP2 accessibility.

Our nucleotide-resolution maps also reveal the influence that primary DNA sequence has on VP16-induced Top2 activity in vivo. Subtle periodic skews in base composition flanking the CC site, similar but positioned differently to that of *E. coli* DNA gyrase[62], are consistent with how Top2 is known to bend the

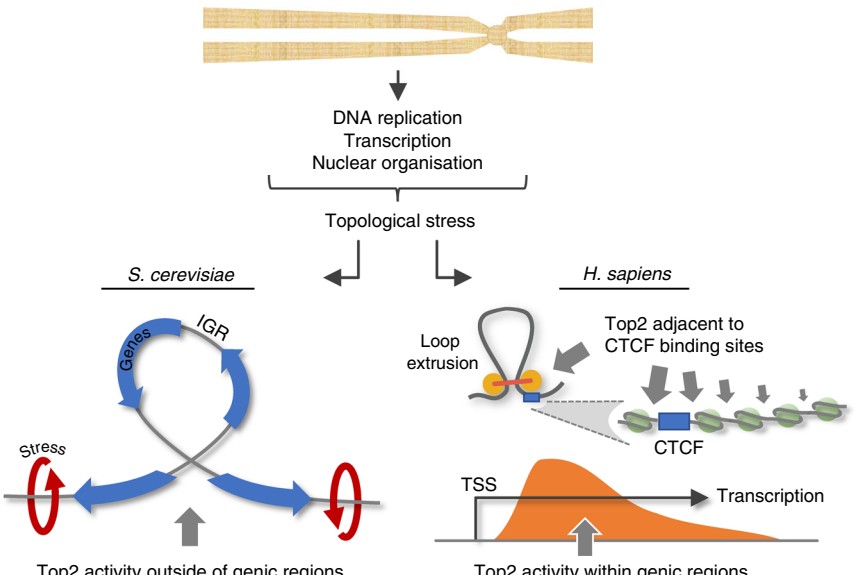

**Fig. 7** Comparison of Top2 activity in *S. cerevisiae* and human cells. Topological stress arising within chromosomes is primarily resolved by Top2 in IGRs in the compact *S. cerevisiae* genome. By contrast, in human cells, TOP2 activity is located around the binding sites of CTCF—a nuclear organising factor that both patterns cohesin dependent loop extrusion and causes local nucleosomal ordering—and also preferentially towards the 5′ end of genes, where it correlates with local transcriptional activity

DNA helix during catalysis[63,64]. Additionally, strong skews towards cytosine in the base immediately 5′ to the scissile phosphodiester bond, in both yeast and human maps, corroborate prior in vitro studies attributing preferential Top2 trapping at cytosines to base-stacking interactions of VP16 with guanosine on the non-scissile strand[52,53,56]. Moreover, the fact that we identify many genomic sites with differential cleavage on Watson vs Crick strands, and that such locations display skews towards the presence or absence of a cytosine 5′ to the scissile bond, lead us to the conclusion that local DNA sequence directly influences the probability of Top2 CC-DSB versus Top2 CC-SSB formation. Importantly, our inference that CC-seq is able to map Top2 CC-SSBs poses the interesting possibility that even sites with no strand disparity (cognates) may be stochastic sites of SSB formation, perhaps driven by independent poisoning of the Top2 active sites by VP16.

We have also used CC-seq to probe MRX-dependent repair of Top2 CCs in *S. cerevisiae*. Using an internal human standard to calibrate signal intensity enabled the relative quantification of Top2 CCs, and thus directly demonstrated that loss of Mre11 activity increases cellular levels of Top2 CCs, corroborating and extending prior genetic experiments in this organism[32,65], and others[35,36]. The antigen-independent enrichment step allows calibration to be performed without genetic epitope-tagging or reconstitution of recombinant protein-DNA complexes, making CC-seq suited to the future study of repair dynamics.

Looking more broadly, we have demonstrated that CC-seq maps CCs of both human TOP2α and TOP2β—proteins that are conserved amongst all vertebrates and whose functional differences remain a subject of interest within the field[66]. Given the essential requirement for TOP2α during mitosis[12], and the apparent inability of TOP2β to compensate for its loss[67], it will be particularly interesting to determine the genomic distribution of TOP2α/TOP2β activities throughout the cell cycle, and to relate this to 3D chromatin conformation measured by Hi-C. Whilst not essential for cell proliferation, TOP2β plays an important role in promoting transcriptional programmes associated with neuronal development[13], and this function cannot be supported by TOP2α. Furthermore, loss of TDP2 function inhibits TOP2-

dependent gene transcription and leads to neurological symptoms including intellectual disability, seizures and ataxia[14]. Thus, accurate genomic maps of TOP2β CC formation in these tissues will help to identify those genes most pertinent to the development of neurodegeneration. Moreover, as demonstrated by our Spo11 maps, the generalised principles of CC-seq are applicable for mapping not just Top2 activity in diverse organisms, but also for mapping similar types of topoisomerase-DNA CCs such as DNA gyrase, Top1 and Top3.

## Methods

**Yeast strains, culture methods and treatment.** The *Saccharomyces cerevisiae* yeast strains used in this study are described in Supplementary Table 3 and were derived using standard genetic techniques. Strains used for Spo11 mapping are isogenic to the SK1 subtype and carry the sae2Δ::kanMX gene disruption allele. Strains used for Top2 mapping are isogenic to the BY4741 subtype and carry the *pdr1DBD-CYC8* drug sensitivity cassette[32]. For Spo11 DSB mapping, cells were induced to undergo synchronous meiosis as follows: Cells were grown overnight to saturation in 4 ml YPD medium (1% yeast extract, 2% peptone, 2% glucose supplemented with 0.5 mM adenine and 0.4 mM uracil) at 30 °C, then diluted to $OD_{600}$ 0.2 in 200 ml YPA medium (1% yeast extract, 2% peptone, 1% potassium acetate) and grown vigorously for 15 h at 30 °C. Cells were then washed with water, resuspended in 200 ml sporulation medium (2% potassium acetate supplemented with diluted amino acids), incubated vigorously for 6 h at 30 °C and harvested by centrifugation. For VP16 treatment, cells were grown overnight to saturation in 4 ml YPD medium at 30 °C, then diluted to $OD_{600}$ 0.5 in 100 ml YPD and grown until $OD_{600}$ 2. 50 ml cultures were then incubated for a further 4 h in the presence of either 1 mM VP16 or 2% DMSO, before harvesting by centrifugation.

**Human cell lines, culture methods and treatment.** The human cell lines used in this study are described (Supplementary Table 4). Human hTERT RPE-1 cells (normal diploid karyotype, immortalised non-cancer) were obtained from ATCC and cultured at 37 °C, 5% $CO_2$ and 3% $O_2$ in Dulbecco's Modified Eagle's Medium DMEM/F-12 (Sigma), supplemented with 10% Fetal Calf Serum (FCS) and 90 Units/ml Penicillin-Streptomycin. For all experiments (CC-seq, Slot Blot, WB, IF, FACs) with asynchronous cell populations, RPE-1 cells were seeded at a density of $(3.5 \times 10^3$ cells/cm²l) and incubated for 72 h at 37 °C, to ensure subconfluent log-phase growth (~70% confluency) at the time of the experiment. For G1 cell populations, WT and *TOP2B*$^{-/-}$ RPE-1 cells were seeded at a density of $5 \times 10^3$ cells/cm² and incubated for 48 h at 37 °C in DMEM/F12 containing 10% FCS, prior to a further 24 h incubation in DMEM/F12 medium containing no FCS to ensure complete G1-phase arrest (verified by FACs, see below). For experiments with proteasome inhibitor and VP16 (CC-seq, Slot Blot), cells were preincubated with 5 μM MG132 (Sigma) for 90 min, trypsinised and incubated in suspension with

5 μM MG132 and 100 μM VP16 (Sigma) for 20 min at 37 °C. For experiments with VP16 alone (IF), adherent cells were treated with 100 μM VP16 for 20 min at 37 °C.

**Generation of *TOP2B*$^{-/-}$ RPE-1 cells.** The oligonucleotides 5′-CACCGCCGCAG CCACCCGACT and 5′-AAACAGTCGGGTGGCTGCGGC (identified using Benchling; https://benchling.com) were annealed and cloned into pX330 following *Bbs*I restriction, as described previously[68]. This SpCas9/ trugRNA co-expression plasmid was transiently expressed in RPE-1 cells to target the 17 bp target sequence GCCGCAGCCACCCGACT (TGG) within exon 1 of *TOP2B*. Single clones were trypsinised and passaged to isolated culture vessels prior to screening for absent protein by Western Blot (WB). All experiments involving *TOP2B*$^{-/-}$ were conducted using the RPE-1 clone T2B/6, in which no TOP2β is detectable by WB or IF.

**Spot tests of chronic VP16 sensitivity.** Single colonies of each *S. cerevisiae* strain were incubated overnight in 4 ml YPD at 30 °C with shaking. 0.1 ml of this starter was used to inoculate 4 ml YPD, prior to incubation for 5 h at 30 °C. Cultures were diluted to make a stock with an $OD_{600}$ of 2.0, then this was 10-fold serially diluted five times. Each dilution was spotted onto plates containing 0, 0.1, 0.3 or 1.0 mM VP16, prior to incubation for three days at 30 °C. Plates were imaged on a 2400 Photo scanner (Epson).

**Southern blotting of meiotic Spo11 DSBs.** Genomic DNA (isolated by non-proteolysing Phenol-Chloroform extraction, as described below) was digested at 37 °C overnight using *Pst*I restriction enzyme (NEB) in NEBuffer 3.1 (100 mM NaCl, 50 mM Tris Base·HCl pH 7.9, 10 mM $MgCl_2$, 100 μg ml-1 BSA). Additional *Pst*I was added for 4 h. Samples were split and ~20 μg subjected to silica-column based enrichment and elution as described below but without prior sonication. Input, flow-through, washes, and eluted samples were then mixed with NEB purple loading dye to 1×, and proteolysed using 1 mg ml$^{-1}$ Proteinase K (Sigma) at 60 °C for 30 min, left to reach room temperature before loading on a 0.7% 1× TAE agarose gel (40 mM Tris Base·HCl, 20 mM glacial acetic acid, 1 mM EDTA pH 8.0) containing 50 μg ml-1 ethidium bromide. Input, flowthrough, and washes contain ~1/10th (2 μg equivalent) of input DNA material. Elution lane contains ~5x greater loading, equivalent to ~10 μg of input DNA. DNA was separated in 1× TAE at 60 V for 18 h. The gel was imaged using InGenius (Syngene) bioimaging system to check migration and then exposed to 180 mJ/m$^2$ UV in the Stratalinker (Stratagene). The gel was then soaked in three times its volume of denaturation solution (0.5 M NaOH, 1.5 M NaCl) for 30 min and then transferred to Zetaprobe (Bio-Rad) membrane by means of a vacuum at 55 mBar for 2 h. After transfer the membrane was washed in water ten times and then cross-linked by exposing the membrane to 120 mJ/m$^2$ UV in the Stratalinker. The membrane was incubated in 30 ml of hybridisation buffer (0.5 M NaHPO$_4$ buffer pH 7.5, 7% SDS, 1 mM EDTA, 1% BSA) at 65 °C for 1 h. The *MXR2* probe for looking at the *HIS4::LEU2* locus was created from 50 ng of template DNA, 0.1 ng of Lambda DNA (NEB) digested with *Bst*EII (NEB), and water. The mix was denatured at 100 °C for 5 min then put on ice. High Prime (Roche) was added in addition to 0.5–3 mBq of α-$^{32}$P dCTP and incubated at 37 °C for 15 min. 30 μl 1× TE was added and the probe spun through a G-50 spin column (GE Healthcare) at 400 × *g* for 2 min. The probe was then denatured by incubating at 100 °C for 5 min and then put on ice before being added to 20 ml hybridisation mixture. The original 30 ml hybridisation buffer was discarded and the 20 ml containing the probe was added to the membrane and incubated overnight at 65 °C. After incubation, the membrane was washed five times with 100 ml pre-warmed Southern wash buffer (1% SDS, 40 mM NaHPO$_4$ buffer pH 7.5, 1 mM EDTA) and exposed to phosphor screen overnight and imaged using Fuji FLA5100, and ImageGauge software (Fuji, version 4.1).

**Western blotting.** Whole human cell extracts (WCE) were harvested by direct lysis in 1× Laemmli loading buffer, denatured for 10 min at 95 °C and sonicated for 30 s using Bioruptor® Pico. Samples were subjected to SDS-PAGE (7% or gradient gel) and transferred to nitrocellulose membrane. Primary immunodetection was by overnight incubation at 4 °C with antibodies targeting TOP2β (Clone 40, BD Biosciences), TOP2α (ab52934, Abcam), or Ku80 (ab80592, Abcam), all at 1/1000 dilution. Secondary immunodetection was by incubation for 1 h at room temperature with a 1/10,000 dilution of HRP-conjugated Rabbit anti-Mouse IgG (ThermoFisher), prior to detection of peroxidase activity using ECL reagent and X-Ray film (Scientific Laboratory Supplies Ltd). Uncropped blot images are present in the supplementary data file.

**Slot blotting.** Samples were diluted fourfold (500 μL total volume) in NaPO$_4$ buffer (25 mM, pH 6.5), and slot blotted onto 0.2 μM nitrocellulose membrane (Amersham), using the Minifold I (Whatman) manifold. The wells were washed twice with 750 μL NaPO$_4$ buffer. The membrane was then blocked with 10% milk-TBST for 1 h at room temperature, prior to incubation overnight with 1/1000 anti-TOP2β antibody (Clone 40, BD Biosciences) at 4 °C. The membrane was washed four times with TBST, incubated with a 1/10000 dilution of HRP-conjugated Rabbit anti-Mouse IgG (ThermoFisher) for 1 h at room temperature, washed four times with TBST, and incubated with ECL detection reagent for 1 min. X-Ray film was used for detection.

**Fluorescence-assisted cell sorting (FACS).** Approximately 10 million RPE-1 cells were trypsinised, washed once in PBS and resuspended in 1.5 ml PBS. 3.5 ml ethanol was added dropwise, with vortexing. Cells were fixed for 1 h at 4 °C, prior to centrifugation and aspiration of the supernatant. Cells were washed twice with PBS, prior to resuspension in 0.5 ml 0.25% Triton-X100-PBS for 15 min on ice. Cells were pelleted by centrifugation, supernatant was aspirated, and the pellet was resuspended in 0.5 ml TBS containing 10 μg/ml RNase A (Sigma) and 167 nM Sytox Green (ThermoFisher). After 30 min incubation in the dark at room temperature, the suspension was filtered through fine mesh into test tubes. For each condition, DNA content in >70,000 cells was analysed using the Accuri C6 (BD Biosciences), with gating based on SSC-A/SSC-H to exclude doublets and cell debris.

**Immunofluorescence.** Cells were seeded onto glass coverslips (Agar Scientific), incubated and treated according to the protocol outlined above. Cells were then fixed with 4% paraformaldehyde PBS for 10 min, washed three times with PBS, permeabilised with 0.2% Triton-X100-PBS for 10 min, blocked for 1 h with 10% FCS-PBS, incubated for 1 h with 1/500 dilutions of primary antibodies targeting Phospho-Histone H2AX (S139) (JBW-301, Merck Millipore) and TOP2α (ab52934, Abcam), washed three times with PBS, incubated with 1/1000 dilutions of secondary antibodies Alexa 488-conjugated Goat anti-mouse IgG (Fisher) and Alexa 647-conjugated Goat anti-Rabbit (Fisher), washed three times with PBS, washed once with distilled water, and mounted with VECTASHIELD containing DAPI (Vector Laboratories).

**High-content microscopy.** Automated wide-field microscopy was performed on an Olympus ScanR system (motorised IX83 microscope) with ScanR Image Acquisition and Analysis Software, 40×/0.6 (LUCPLFLN 40× PH) dry objectives and Hamamatsu ORCA-R2 digital CCD camera C10600. Numbers of anti-phospho-Histone H2AX (S139) foci (Alexa 488; FITC filter) were quantified in the nuclear region colocalising with DAPI, after first gating for cells based on area and circularity factor, using Olympus ScanR Analysis software.

**Isolation of nonproteolysed DNA from yeast and human cells.** For generation of CC-seq libraries from human cells, $1.5 \times 10^7$ RPE-1 cells were treated as described above, pelleted by centrifugation, washed once with 15 ml ice-cold PBS, and resuspended in three aliquots of 400 μL ice-cold PBS. For generation of CC-seq libraries from *S. cerevisiae* cells, $1 \times 10^9$ yeast cells were treated as described above, then spheroplasted in 1.5 ml spheroplasting buffer (1 M sorbitol, 50 mM NaHPO$_4$ buffer pH 7.2, 10 mM EDTA) containing 200 μg/ml Zymolyase 100 T (AMS Bio-tech) and 1% β-mercaptoethanol (Sigma) for 20 min at 37 °C. Two microlitres of protease inhibitor cocktail and 2 μL Pefabloc (Sigma) were added before splitting into five aliquots of 400 μL. All subsequent steps of the protocol are the same for yeast and human samples. One millilitre of ice-cold ethanol was added to 400 μL cell suspensions in microcentrifuge tubes in order to rapidly denature proteins in situ, mixed, incubated for 10 min on ice, and pelleted by centrifugation. The supernatant was thoroughly removed by aspiration, prior to addition of 200 μL 1× STE buffer (2% SDS, 0.5 M Tris pH 8.1, 10 mM EDTA, 0.05% bromophenol blue), cell disruption using a pestle (VWR), addition of a further 400 μL 1× STE buffer, and incubation for 10 min at 65 °C. Samples were cooled on ice and 500 μL Phenol-Chloroform-isoamyl alcohol (25:24:1; Sigma) was added. The mixtures were emulsified by shaking and pipetting 5 times with a 1 ml micropipette, prior to phase separation by centrifugation at 20,000 × *g* for 20 min. By minimising mechanical shearing of the lysate prior to phenol chloroform extraction, peptides that are covalently linked to high molecular weight DNA segregate into the aqueous phase alongside other purified nucleic acids. Free protein, coated in SDS, will partition to the interphase. Five hundred microlitres of the aqueous phase was removed to a clean microcentrifuge tube, taking care not to disturb the interphase, and nucleic acids were precipitated with 1 ml ice-cold ethanol, pelleted by centrifugation, washed with ice-cold 70% ethanol, and dissolved in TE buffer overnight at 4 °C. Samples were then incubated with 0.2 mg/ml RNase A (Sigma) for 1 h at 37 °C; nucleic acids were precipitated with 1 ml ethanol, pelleted by centrifugation, washed twice with 70% ethanol and dissolved in TE overnight.

**Enrichment of covalent protein-linked DNA (CCs).** Aliquots of nonproteolysed DNA (prepared as above) were combined to a total of 1 ml and sonicated to an average fragment size of 300–400 bp with Covaris (duty cycle: 10%, intensity/peak power incidence: 75 W, cycles/burst: 200, time: 15 min). One millilitre of sonicated sample was diluted to reduce viscosity by addition of 1 ml TE. Triton-X100, N-Lauroylsarcosine sodium salt and NaCl were then added to complete the binding buffer (final concentrations: 0.3 M NaCl, 0.2% Triton-X100, 0.1% N-Lauroylsarcosine sodium salt). Each sample was divided over several Miniprep (QIAGEN) silica-fibre membrane spin columns, such that the total DNA loaded to each was ~20 μg. Under these high salt conditions, protein, but not nucleic acids, bind to the silica membrane, leading to selective retention of any CCs. The flow-through was reapplied to the column to improve yield. Columns were washed six times with 600 μL of TEN (10 mM Tris, 1 mM EDTA, 0.3 M NaCl) per 1 min wash to remove any residual non-CC DNA fragments, prior to elution with 100 μL TES

(10 mM Tris, 1 mM EDTA, 0.5% SDS). The SDS detergent in the elution buffer releases interactions between protein and silica, thereby releasing bound CCs.

**DNA end repair and adapter ligation.** Eluted products were pooled to 500 μL in TES and incubated with 1 mg/ml Proteinase K (Sigma) for 30 min at 60 °C, prior to overnight ethanol precipitation at −80 °C with 1.41 ml ethanol, 0.2 mg/ml glycogen and 200 mM NaOAc. The DNA-glycogen precipitate was pelleted by centrifugation at 20,000 × g for 1 h at 4 °C, washed once with 1.5 ml 70% ethanol, and re-pelleted by centrifugation. The supernatant was aspirated, and the pellet was air-dried for 10 min at room temperature, prior to solubilisation in 52 μL 10 mM Tris-HCl. DNA concentration was measured in a 2 μL sample with the Qubit (ThermoFisher) and High Sensitivity reagents. The remaining 50 μL was used as input for one round of end repair and adapter ligation with NEBNext Ultra II DNA Library Preparation kit (NEB), according to manufacturer's instructions, except for the use of a custom P7 adapter (Supplementary Table 5). It is our understanding that NEBNext Ultra II contains a polymerase capable of blunting ends by either extending the complementary strand towards a ssDNA 5′ extension, or exonucleolytically trimming a ssDNA 3′ extension, but that ssDNA 5′ extensions are not degraded. This kit also provides 3′ dATP terminal transferase activity enabling ligation of adapters containing a terminal 3′ dTTP. The use of custom adapters is to allow differentiation of the sheared end (P7 adapter) from the Top2/Spo11 end (P5 adapter). After ligation of the P7 adapter, DNA was isolated with AMPure XP beads (Beckman Coulter) according to manufacturer's instructions (beads:input of 78:90) and eluted in 50 μL 10 mM Tris-HCl. Samples were diluted twofold with 50 μL TDP2 reaction buffer (100 mM TrisOAc, 100 mM NaOAc, 2 mM MgOAc, 2 mM DTT, 200 μg/ml BSA) and incubated with 3 μL of 10 μM recombinant human TDP2[26,69], for 1 h at 37 °C. DNA was isolated again with AMPure XP beads (beads:input of 103:103) and eluted in 52 μL 10 mM Tris-HCl. Next, a second round of adapter ligation was conducted using adapter P5, without prior end repair in order to prevent dephosphorylation of DNA 3′-phosphate termini which may be generated by TDP2 activity on 3′-CCs. After ligation of the P5 adapter to the Top2/Spo11-cleaved end, DNA was isolated with AMPure XP beads (beads:input of 78:90) and eluted in 17 μL 10 mM Tris-HCl.

**PCR and size selection of CC-seq libraries.** DNA concentration was measured in 2 μL using the Qubit. The remaining 15 μL was used as template for the PCR step of the NEBNext Ultra II PCR step using universal primer (P5 end) and indexed primers for multiplexing (P7 end), according to manufacturer's instructions. PCR reactions were diluted with 50 μL 10 mM Tris-HCl, DNA was isolated with AMPure XP beads (beads:input of 84:100) and eluted in 30 μL 1 mM Tris-HCl pH 8.1. Samples were then subjected to 200–600 bp size selection using the BluePippin (Sage Science), prior to quantification of molarity using the Bioanalyzer (Agilent).

**Deep sequencing and data analysis of CC-seq libraries.** Multiplexed library pools were sequenced on the Illumina MiSeq (Kit v3–150 cycles) or Illumina NextSeq 500 (Kit v2–75 cycles), with paired-end read lengths of 75 or 42 bp, respectively. Paired end reads that passed filter were aligned using bowtie2 (options: -X 1000 –no-discordant –very-sensitive –mp 5,1 –np 0), using MAPQ0 settings for yeast or MAPQ10 settings for human experiments, then SAM files processed via terminalMapper (Perl, v5.22.1; https://github.com/Neale-Lab) that computes the coordinates of the protein-linked 5′-terminal nucleotide. The reference genomes used in this study are hg19 (human), and Cer3H4L2 (S. cerevisiae), which we generated by inclusion of the his4::LEU2 and leu2::hisG loci into the Cer3 yeast genome build. Yeast data sets were filtered to exclude long terminal repeats, retrotransposons, telomeres, and the rDNA. Human datasets were filtered to remove known ultra-high signal regions[70,71] and repeat regions[71]. All subsequent analyses were performed in R (Version 3.5) using RStudio (Version 1.1.383), unless indicated otherwise.

**Detailed description of terminalMapper functionality.** Tab-delimited.SAM files, the primary output of Bowtie2 alignment, specify (i) 1-based leftmost coordinates for mapped reads (ii) numerical "flags" denoting the aligned identity of each read pair—terminalMapper only handles paired, fully aligned reads (i.e. 99—Read-1, Watson | 147—Read-2, Crick//83—Read-1 Crick | 163—Read-2, Watson) (iii) Alpha-numeric CIGAR codes describing base-by-base alignment and detailing the presence of INDELs—terminalMapper utilises such information for accurate coordinate calling. For example, 5M2I30M1D25M denotes:

- 5 bp reference match (5 M) ("M" may contain unspecified mismatches)
- 2 bp insertion in the read (relative to the reference) (2I)
- 30 bp reference match (30 M)
- 1 bp deletion in the read (relative to the reference) (1D)
- 25 bp reference match (25 M)

(iv) Alpha-numeric MD:Z tags denoting the position and base composition of SNPs/deletions present in the read relative to the reference. Insertions are not specified. terminalMapper utilises MD:Z tags to detect and discard reads with ambiguous ends. In LIBRARY_TYPE = SINGLE mode, ≥2 bp of mismatch at the Read-1 5′-end is defined as ambiguous and disqualifies the read pair from the main dataset. In LIBRARY_TYPE = DOUBLE mode, ≥2 bp of mismatch at either the

Read-1 5′ or Read-2 5′ end disqualifies the read pair. Non-informative 3′-ends are not considered. For example, an MD:Z-tag of 0T0C25A5^T10 denotes:

- An initial 2 bp mismatch (reference specifies TC, read contains alternative bases) (0 T, 0 C)
- 25 bp of precise reference:read match (25) followed by a 1 bp mismatch (25 A)
- 5 bp of precise reference:read match (5) followed by a 1 bp deletion in the read (reference contains a T) (5^T)
- 10 bp of precise reference:read match (10)

ATGAGCGTACCTGTAAATAAGAAGATCGATCGA_GGTACACATACT — READ (0T0C25A5^T10)

TCGAGCGTACCTGTAAATAAGAAGATCAATCGATGGTACACATACT — REFERENCE

For unambiguous 99–147 and 83–163 read pairs, coordinate positions of the informative ends are calculated. As SAM files specify 1-based leftmost coordinates—the 5′ end is readily called by Bowtie2 for Watson (+) reads (99 or 163). In contrast, for Crick (−) reads (83 and 163), the leftmost base is the 3′ end of the read. To call 5′ Crick (−) coordinates, CIGAR codes are parsed and scored to determine the mapped read length—according to the following rules: (M = 1, D = 1, I = 0)—and the sum is added to the 3′ coordinate. Insertions (I) (in the read) are ignored in order to call coordinates accurate to the utilised reference. As an example, a Crick (−) read with a CIGAR code of 75 M and a leftmost coordinate is 10200 is called as 102074 (10200 + 75−1). A 1 bp adjustment is made as the leftmost base is included as part of 75 M. A more complex Crick (−) read with a CIGAR code of 35M2D10M3I30M and a leftmost coordinate of 10200 is called as 10276 (10200 + 35 + 2 + 10 + 30 −1).

**Calibrated CC-seq library generation.** Calibration of Top2 CC-seq experiments was conducted to allow comparison of relative peak intensities in different yeast strains. This was achieved by spike-in of human DNA following the sonication stage of the protocol, which is a method that has been used to calibrate other sequencing methods[72,73]. S. cerevisiae and RPE-1 cells were exposed to VP16 and processed until just after sonication, exactly as according to the cell treatment and CC-seq protocols above. DNA concentration was quantified by Qubit, and then mixed at a molar ratio of human DNA:yeast DNA of 1:100. All subsequent stages of the protocol were identical, except read alignment, for which we used both hg19 and Cer3H4L2 builds successively. Cer3H4L2-aligned peak heights were corrected in each sample by multiplying by the reciprocal fraction of human reads in that sample.

**Fine-scale plotting of CC-seq libraries.** Fine-scale (nucleotide resolution) maps of Spo11/Top2 CCs were produced as simple histograms over a specified region. Dark red and blue line heights indicate numbers of 5′-terminal nucleotides detected at that position on the Watson and Crick strands, in units of HpM. Where indicated in the figure caption, smoothed data are also plotted as pale red and blue polygons, in addition to the unsmoothed nucleotide resolution data. This smoothing was either applied using a sliding Hann window of the indicated width, or using a custom smoothing function (VarX), as indicated in the figure caption.

**Broad-scale plotting of CC-seq libraries.** Broad-scale maps of Top2 CCs were produced by binning nucleotide resolution data at 10 or 100 Kbp resolution. Binned data were either plotted directly; or first scaled according to the estimated noise fraction (see below), smoothed with a 10-bin Hanning window, and median subtracted.

**Estimation of the noise fraction in human CC-seq libraries.** The noise fraction in each sample was estimated using an adaptation of the previously published NCIS method[74]. Briefly: the data for −VP16 and +VP16 samples were first binned at 10 Kbp resolution. Then the subpopulation of bins with the lowest TOP2 CC signal was identified in each sample. The average signal density of this subpopulation of bins, in each sample, was defined as the noise density ($d_{-VP16}$ and $d_{+VP16}$). Signal in the -VP16 sample was scaled by a normalisation factor defined by Eq. (1).

$$r = \frac{d_{+VP16}}{d_{-VP16}} = 0.736 \qquad (1)$$

**Genomic loci of interest used in this study.** Published datasets employed in this study are listed in Supplementary Table 6. In both yeast and human, TSSs were defined using recent nucleotide resolution maps of transcription initiation (CTSS) defined by Cap Analysis of Gene Expression (CAGE[39,49]). Loci were stratified based on associated gene length, expression level in vegetative SK1[40] (GSM907178, GSM907179, GSM907180), or expression level in meiotic SK1[40] (GSM907176, GSM907177). Human loci were stratified based on associated gene length, or expression level[48] (GSM1395252, GSM1395253, GSM1395254). Occupied RPE-1 CTCF motifs were identified as follows: The FIMO tool[75] and the CTCF Position Weight Matrix (PWM) from the JASPAR database[76] were used to find all significant hg19 CTCF motifs ($p < 1 \times 10^{-4}$). These were filtered to include only those which overlapped positions of RPE-1 CTCF ChIP-seq peaks[71] (GSM749673, GSM1022665), and stratified based on this ChIP-

seq data. RPE-1 loop anchor-associated CTCF motifs were identified using the Juicer MotifFinder[77] with the RPE-1 WT Hi-C looplist[47] (GSE71831) and CTCF ChIP-seq BED file[71] (GSM749673, GSM1022665) as input. Human TOP2 CC-seq peak coordinates were identified by thresholding (0.05 HpM) of pooled +VP16 data on Chr1, as a representation of the global pattern.

**Aggregation of CC-seq data around loci of interest**. Nucleotide resolution CC-seq data (line-plots) or binned CC-seq data (heatmaps) were aggregated within regions of specified size, centred on the loci of interest. The resulting sum total HpM was divided by numbers of loci to give a mean HpM per locus.

**Quantification of Spo11 and Top2 activity in defined regions**. Yeast Spo11 hotspots were the same as defined previously[28]. Yeast intra/intergenomic regions were defined based on start and stop coordinates reported on the Saccharomyces Genome Database (https://www.yeastgenome.org). Nucleotide resolution CC-seq data were tallied within defined regions and are expressed as an aggregate (barplot), showing percentage of total signal, and as the distribution of signal densities (box-and-whisker plot), as indicated in figure captions. Human chromatin compartments A and B were defined based on eigenvector analysis of previously published 100 kb resolution RPE-1 Hi-C data[47] using the Juicer package[43,77]. CC-seq data, binned at 50 bp resolution (+VP16 condition), were tallied within defined regions, and are expressed as the distribution of signal densities (box-and-whisker plot), as indicated in figure captions. For analysis of TSS-proximal and CTCF-proximal regions, CC-seq data, binned at 50 bp resolution (+VP16 condition), were tallied within regions 5 kb upstream or downstream of the TSS, or within the 100 bp region centred on CTCF-bound CTCF motifs. Tallied CTCF-proximal CC-seq signals were plotted against colocalizing ChIP-seq signal[71]. Tallied CC-seq signals downstream of the TSS were plotted against gene expression level[48], Transcript initiation frequency[49], or GRO-seq signal[50].

**Correlation between Watson and Crick cleavage positions**. Nucleotide resolution human TOP2 data was thresholded at 0.01 HpM. Nucleotide resolution yeast Top2 and Spo11 data were not thresholded. Peak coordinates on the Crick strand were offset over the range of −100 to +100, relative to Watson coordinates. After each offset, the data was filtered to include only sites with both Watson and Crick hits. The Pearson correlation (r) between the Watson and Crick signal intensities in these $n$ sites was calculated. We also counted the fraction of reads found within these $n$ sites, and normalised this number over the −100 to +100 bp range. Data are expressed as Pearson r values and/or normalised HpM values, as indicated in the figure captions.

**Spatial correlation of *S. cerevisiae* Top2 and Spo11 signals**. Nucleotide resolution Top2 and/or Spo11 CC-seq data was binned at 50 bp resolution and the Pearson correlation was calculated between the sum HpM for pairs of bins separated by increasing inter-bin distance.

**Fractionation of cognate and noncognate Top2 CC sites**. Nucleotide resolution Top2 CC-seq data on the Watson strand were offset by 3 bp relative to data on the Crick strand. 3 bp offset sites were tested for significant non-equivalence using the Poisson exact test. Sites with fewer than 8 hits in total were discarded, due to low statistical power. Sites with a P value of ≤0.05 were defined as noncognates, sites with a P value > 0.05 were defined as cognates, and sites with a P value > 0.95 were defined as highly cognate. A randomisation experiment was conducted in order to estimate expected numbers of cognates and noncognates within the sample under a random model where signal is distributed independently on each strand. To achieve this, the amplitudes of Top2 CCs (HpM) were shuffled amongst the positions in the nucleotide resolution datasets, prior to fractionating as described above.

**Simulations of cognate Top2 CC sites**. Observed and randomised data were compared to a simulation of 100% cognate sites. First we averaged observed CC-seq data on the Watson and Crick strand at every position to create a vector of $n$ lambda values. To create the simulated data on the Watson strand we then sampled once from each of these $n$ Poisson distributions. The process was repeated to generate the simulated data on the Crick strand. We subsequently fractionated this simulated data using the same process used for the observed data.

**DNA sequence composition of cognate and noncognate Top2 CCs**. Nucleotide resolution Human and Yeast Top2 CC-seq data were first fractionated into cognate and noncognate sites as described above. Data were then normalised such that the total CC-seq signal (Watson + Crick) at each site was equal, without changing the Watson:Crick ratio at any site, in order to decrease the influence of other features of individual genomic loci unrelated to DNA sequence (for example chromatin accessibility). For example, the two sites W9:C1 and W30:C10 became W90:C10 and W75:C25, respectively. For cognate Top2 CCs, a dyad axis coordinate was defined as the centre point between the Top2-linked nucleotides on the W and C strand (that is, the midpoint of the central two base pairs in the four base pair overhang). For noncognate Top2 CCs, we used an inferred dyad axis in the

same relative position. Watson and Crick DNA sequence orientated 5′−3′ was aggregated ±20 bp around these two classes of Top2 CCs, weighted by the normalised Watson and Crick CC-seq values, using seqBias (Perl, v5.22.1; https://github.com/Neale-Lab). This method is designed to ensure that DNA sequences around CC-seq sites are sampled proportionally to the relative degree of disparity. Statistical significance was determined using the one sample (goodness-of-fit) Chi-squared test, as described previously[52]. Data are presented as fractional base composition (line plots), or as Log2 fold deviations from local average (yellow-red heatmap), or Log2 fold difference between cognate and noncognate (white-black heatmap).

**Ideograms**. Human chromosome ideograms were adapted from the open source ideogram package (https://eweitz.github.io/ideogram/).

**Reporting summary**. Further information on research design is available in the Nature Research Reporting Summary linked to this article.

## Data availability

The source data underlying Figs. 1b and 3a and Supplementary Figs. S7a–d are provided as a Source Data file. CC-seq data used in Figs. 1b–d, 2b–f, 3b–f, 4a–f, 5a–j, 6a–f and Supplementary Figs. 2a–e, 3a–k, 4d–e, 5a–j, 6a–e, 7e, 8a–c, 9a–f, 10a–f, 11a–d, 12b–e, and 13d have been deposited in the NCBI GEO database under accession numbers GSE136943 (human Top2), GSE136675 (*S. cerevisiae* Top2) and GSE137685 (*S. cerevisiae* Spo11). Raw unmapped FASTQ data underlying the same figures have been deposited in the NCBI SRA database under the accession numbers SRP186470 (*S. cerevisiae* Spo11), SRP186446 (*S. cerevisiae* Top2), SRP187576 (human Top2). All strains and cell lines listed in Supplementary Tables 3 and 4 are available from the corresponding author upon request.

## Code availability

Code used in our FASTQ-processing pipeline (terminalMapper), and in analysis of local DNA sequence composition (seqBias) is available on our open access Github repository (https://github.com/Neale-Lab).

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

## Acknowledgements

We thank Keith Caldecott, Antony Oliver and Peter Hornyak for sharing recombinant TDP2, and Jon Nitiss for sharing the *pdr1Δ* strains. W.G., D.J., R.A., T.J.C., and M.J.N. were supported by an ERC Consolidator Grant (#311336), the BBSRC (#BB/M010279/1) and the Wellcome Trust (#200843/Z/16/Z).

## Author contributions

W.G., D.J. and M.J.N. conceived the project and developed the CC-seq methodology. W.G., D.J., H.T. and R.A. generated whole genome CC-seq libraries. W.G. performed all data processing and analysis, and all human cell work. D.J., H.T. and R.A. performed yeast sensitivity and meiotic DSB assays. T.J.C. developed the mapping pipeline and associated tools. W.G. and M.J.N. interpreted the observations and wrote the manuscript.

## Competing interests

The authors declare no competing interests.
