## [Peer Review File · Nature Communications]

Reviewers' comments:

Reviewer #1 (Remarks to the Author):

This intriguing paper describes a novel approach ("CC-seq") to mapping covalent protein-DNA linkages in a highly selective and single nucleotide resolution manner. The approach is shown to be generally applicable to mapping of double-strand breaks (DSBs) and (possibly) single-strand breaks (SSBs) made by Spo11 during meiosis and by topoisomerase II (Top2) in response to poisoning by etoposide. The Top2 covalent complexes are mapped in yeast and in cultured human cells. The approach is very clever and appears to be a substantial improvement over existing methods. The data show clearly that the authors have succeeded in developing a technically straightforward, reproducible, specific, and highly sensitive method for enriching and sequencing Spo11 and Top2 covalent complexes. The analysis of the data mostly confirm and extend known patterns, but there are some surprising and interesting new observations that emerge from the high spatial resolution of the data. The datasets will be a valuable resource, but more importantly, the CC-seq method is likely to be powerful new approach to studying this important class of DNA lesion. As such, this manuscript is likely to be of considerable general interest.

Although the paper is mostly well written, there are a number of areas where clarity could be improved or where interpretations could be strengthened. These are all easily addressable by changes to text and figures, with fairly minimal additional data analysis needed. No further experimentation is necessary.

Major points

1. Fig 1A: it would be helpful to provide a cartoon to show how the sequencing library is prepared, especially to allow the reader to understand how DSB and SSB fragments are recovered and sequenced. It would also be helpful to provide more explicit description of the library preparation in the Methods rather than simply referring to the NEB kit used.
2. Line 118: The existence of a base composition bias per se is not evidence of specificity, because library generation or sequencing artifacts would also be expected to have non-random base composition. The pattern needs to be compared to known Spo11 patterns.
3. Lines 152-154: The discussion of the positive correlation of Spo11 activity with gene expression could be developed better. Studies that have probed directly for a causal link have failed to demonstrate that transcription levels dictate DSB levels (e.g., PMIDs 1568254, 26245832). Also, Fig S1D is cited here but S1E is meant.
4. The parts of the manuscript that deal with "DSBs" vs. "SSBs" are difficult to understand and therefore not convincing. One problem is that the text doesn't make a clear distinction between fraction of sites (what is measured) and fraction of breaks. This is a fundamental difference that needs to be more clearly dealt with. One cannot know what fraction of events is SSBs because nicks at "DSB" sites will look like DSBs by this analysis. Example on line 292: "we similarly observe more single-strand Top2 CCs (90.1% and 94.2%, respectively) than double-strand Top2 CCs (9.9% and 5.8%, respectively) following etoposide exposure (Figure 6A and S3C)." This statement is incorrect: The observation here is percent of sites, not percent of CCs. Same concern applies to abstract and other places in the text.
5. More generally, a major problem with the SSB analysis is that operational definitions are never really clearly explained, and the text assumes the interpretation rather than providing a neutral description of the findings first, then providing the interpretation. It would be better to provide a clear explanation of the relevant patterns in the data (namely, that there are many locations that show one-sided Top2CC mapping signal), then to give a discussion of interpretation and alternative possibilities. It will be important to explain how one can exclude systematic biases in recovery, library-prep, or sequencing as the source of the one-sidedness. It will also be useful to explain how the purification and sequencing methodology would be expected to behave when presented with a SSB-Top2CC as opposed to a DSB-Top2CC (see also point 1 above).
6. Fig 6D and lines 322-327: The analysis of methylation was completely uninterpretable as presented, and the conclusions are not convincing. Presentation: Neither the text nor the figure

explain how methylated sites are defined. One has to dig into the methods to find that methylation has not been assayed directly, but rather that methylation data from a separate study were used. Better presentation is needed. Interpretation: to make the conclusion the authors wish to make, they would need to directly assess methylation on Top2-CC fragments. Since that would probably be prohibitively difficult at this point, the use of existing data is warranted, but more care is needed in interpreting the findings, and alternative interpretations should be considered. For example, individual CpG dinucleotides are not always methylated in all cells in a population, so if methylation within the Top2 binding site is the relevant factor, then cells that lack methylation should be competent to make breaks there. Analysis of methylation fraction vs. Top2-CC frequency should be carried out. Also, the authors interpret their findings solely in terms of the effect of the methyl group on Top2 DNA cleavage, but alternative possibilities include occlusion by nucleosomes or other DNA binding proteins; heterochromatin formation and suppression of transcription; possibility that methylated cytosines tend to be within sequence contexts that are poor substrates for Top2; artifacts of Top2-CC recovery or library preparation. Experimental tests of these ideas are possible (e.g., CC-seq in mutants defective for methylation or heterochromatin formation) but are outside the scope of what is necessary for this paper. But absent such data, there are too many caveats, and conclusions that can be drawn from this analysis are limited.

7. The discussion is somewhat repetitive with points made in the introduction and results. The first three paragraphs in particular provide very little new information. It would greatly strengthen the paper if the discussion section were either shortened substantially to remove the redundancy, or rewritten to replace the redundant information with elaboration of points not already covered in detail.

Minor points

8. Please number the pages
9. Fig S1A,B: use consistent labeling (10,100, etc. vs. powers of ten)
10. S1C: text labels are too small
11. Fig. 1E: define "dyad axis"
12. Throughout: the text and figure switch back and forth between "etoposide" and "VP16". It would be better to pick one and stick with it.
13. Fig. 2C Top: it is unclear why the axis is labeled "density" here since the units are still HpM. Is it always "density" but only sometimes stated as such, or is there something special about the way this graph's values are calculated?
14. Fig 3 axis labels are confusing. Some use HpM, others use HpB, and another uses HpBbbp. Since the latter two are not defined in the legend, it was hard to understand what was being presented. More importantly, the switch between units seems arbitrary without an explanation for why it needs to be different in different figures.
15. Fig S5A: what does the lower case "f" mean in "0% FCSf"?
16. Fig S5C: the micrographs can be made larger since this is a supplemental figure
17. Fig S5E: There is room to write out "TOP2B-/-" so no need to introduce a new (undefined) abbreviation.
18. Fig 4A top: What do the red and blue triangles signify? Presumably the motifs, and possibly strand orientation? Please specify in the legend.
19. Fig 6B: Y-axis labels are too small
20. Fig 6B and line 309: The base composition bias is referred to as a "motif" but it is not clear that this term applies here. Was a motif-finding algorithm applied?
21. It is confusing that the first paragraph under the heading on line 289 is about SSBs vs DSBs, which is not mentioned at all in the heading.
22. Line 339: Nucleotide-resolution mapping of unresected Spo11 DSBs was also reported using S1-seq in yeast (PMID: 29523237, 28059759). This could be cited.
23. Line 395: DNA bending by Top2 is well established, so the text here gives a misleading impression that the current findings and the sole cited paper are the first to demonstrate it. See, e.g., PMID: 23580548, 25142513.
24. Lines 401-402: "thus directly demonstrates for the first time that loss of Mre11 activity increases cellular levels of Top2 CCs in *S. cerevisiae*." While formally true as written, this sentence

may be misleading because it is likely to be interpreted that this is the first demonstration in any organism, not just the first demonstration in yeast. It would be better to make it explicit that this has been shown already for cultured human cells (Hoa et al., ref. 32 in this ms.) and *Xenopus* extracts (PMID: 26880199, not cited).

Reviewer #2 (Remarks to the Author):

MAJOR REMARKS

1) Details of the method. In the first paragraph of the Results section and in Fig. 1a, the Authors describe the CC-seq method which they have developed, by adapting to next-generation sequencing (NGS) a previous biochemical strategy used for enriching Spo11 covalent complexes on silica columns. Unfortunately, the description of the method is minimal to an extent that I had to immediately go to the Methods section to find an explanation, and still did not manage to fully understand how the procedure works:

--The first step of the protocol is a standard phenol-chloroform DNA purification procedure, although the initial step in ethanol is not conventional (to my knowledge). What is the purpose of this step?

--The Authors state that "By minimizing mechanical shearing of the lysate prior to phenol chloroform extraction, peptides that are covalently linked to high molecular weight DNA segregate into the aqueous phase". Can the Authors clarify what exactly the aqueous phase will contain after following the procedure they describe? Since there is not a prior protein digestion step (e.g., by proteinase K) and since the cells are fixed in ethanol, I imagine that most of the DNA is actually in chromatin form (coated with histones and other protein complexes) and not only bound covalently to proteins such as Top2.

--After phenol-chloroform, the aqueous phase is passed through a standard miniprep column. In Fig. 1a, the Authors show that this step enriches for CCs between DNA and Top2. How does this work, given the fact that there is no affinity capture? Also, what happens to the other DNA fragments that are presumably coated by proteins (see my comment above), but are not covalently bound to Top2?

--A question related to the one above is that even if we assume that the method selects for CCs, it is possible that the DNA end(s) is(are) linked to a protein different than Top2 (possibly, an unknown enzyme). If the Authors are truly able to biochemically isolate protein-bound (covalently) DNA fragments, they should attempt to perform mass spectrometry to elucidate the nature of the proteins/peptides bound.

--Reading further through the library preparation description in the Methods, I thought I finally understood the trick behind CC-seq, but I am still unsure whether this is truly the case: does it mean that genomic DNA fragments that are not covalently bound to Top2 will be ligated with the P7 adapter on both ends, and thus will not be sequenced, whereas only the fragments bound to Top2, which are end-cleaned by TDP2, can be ligated to the P5 adapter on the end where Top2 was bound, and thus will be sequenced? If this is the case, the Authors should absolutely explain this upfront in the Results section, and also modify Fig. 1a to clearly describe all the steps of the procedure. Also, if this is how CC-seq works, why can't the Authors simply start from purifying DNA over a column followed by the TDP2 step, instead of running through phenol-chloroform?

2) DSBs vs SSBs. In Fig. 6a, the Authors show that most of the signal detected by CC-seq corresponds to SSBs. When I read this I was puzzled, as from my understanding of how the method works I assumed that only DSBs can be captured. Only when reading the corresponding

Methods section did I realize that the Authors come to this conclusion based on their analysis of the genome-wide pattern of reads, i.e., they interpret as SSBs those genomic sites where CC-seq yielded only reads on one or the other strand. Unfortunately, I do not think this is a correct interpretation of the data. As the method is based on the NEBNext kit for library preparation, which uses the T4 ligase to ligate the P5 and P7 adapters to the ends of DNA fragments, it is not at all clear how at this step SSBs could be possibly ligated. Furthermore, if SSBs are present within a genomic DNA fragment, they might simply be sealed by the ligase. What I suspect the Authors are seeing in their data could be either of these two DNA structures:

--Single-ended DSBs that have attached Top2 (novel biology?): for example, at stalled replication forks, if the leading strand breaks, and a single-ended DSB is formed, this might be detected by CC-seq (notably, this end does not necessarily have to be bound to Top2, but could be bound to an unknown protein. This brings me to the suggestion I made above to subject DNA fragments with covalently-bound ends to mass spectrometry).

--DSBs induced by Top2, where only one end has remained covalently bound to Top2, or one end is somehow 'masked' by a large protein complex and did not purify (the second hypothesis less plausible).

The Authors must absolutely clarify these issues and provide a better interpretation of what they name as SSBs are in reality. Towards this goal, it would be useful if the Authors performed an experiment with or without a Top1 inhibitor or another drug causing a dramatic increase in the number of SSBs, and see what CC-seq reveals.

ADDITIONAL REMARKS

1) Although the manuscript is well written, I found that, especially in the Results section, in many occasions the results were only superficially explained and many details were left implicit. This is also the case in figure legends and in plot labels, where many details are left unexplained and abbreviations are often not spelled out (see Minor Comments below). The Authors should strive to provide more details and describe the results in a more explicit manner.

2) Related to the previous point, the Results section often contains interpretations/discussions that should be left for the Discussion section only. Also, the tense occasionally switches from past (narrative style, which is easy to follow) to present, which makes it difficult to follow what is a result and what is more a discussion point. An example would be lines 147 to 154: " [...] the fact that *S. cerevisiae* Top2 CC signal is not correlated [...]. By contrast, Spo11 activity is positively correlated [...].

3) Further related to the previous points, throughout the manuscript I had the feeling that yeast and human results are often mixed. It would be clearer to present all the findings in *S. cerevisiae* first, and then show the results obtained using human cells (the same in the Discussion).

4) It is not clear why the Authors used a human spike-in to normalize yeast libraries and not, for example, a yeast spike-in to normalize human libraries?

5) Line 119: while describing the results shown in Fig. 1e, the Authors should specify how the actual Spo11 binding motif looks like: was this previously known? If not, is the observed sequence preference in line with the structure of DNA-bound Spo11 (if available), as shown for Top2 in Fig. 6?

5) Fig. 3d: the Authors should quantify the correlation between Top2 peaks and CTCF and H3K4Me3 and H3K27Ac peaks, and also provide a quantification of the relative distribution of Top2 peaks in different parts of the genome (coding vs non-coding, exons, introns, TSS, etc).

- 6) The Authors nicely show a strong correlation between DSB maps obtained by End-seq (indirectly) and CC-seq profiles. Even though the Authors provide a clear rationale for developing CC-seq, these methods might be perceived as redundant, in the sense that, in the end, they all show the distribution of DSBs genome-wide (with END-seq and BLESS/BLISS or DSBCapture also detecting DSBs not originating from topoisomerases). The Authors should better contrast these methods in the Discussion, and highlight the advantages of CC-seq over other DSB-detection methods.
- 7) In their CTCF analysis, have the Authors examined whether the loops that have a Top2-CC on their flanks also contain a gene that is actively transcribed? It would be helpful to classify loops based on whether they contain or not a gene (inside the loop or flanking it) and whether the gene is active or not.
- 8) The Authors show that Top2-CC in human cells are abundant within the gene body of transcribed genes, in line with previous findings by other DSB-capturing methods. Can the Authors provide a better quantification of Top2-CCs in gene bodies? Are DSBs occurring all along the gene body or preferentially at the 5' end? Is there a preference for exons, introns or exon-intron boundaries?
- 9) Lines 268-269: how did the Authors come to the conclusion that " [...] Top2 activity is not proportional to local RNA polymerase activity when assayed at finer scale ". Figure 5a actually shows the opposite. Where is the data supporting the Authors' statement?
- 10) Related to point 5), the Authors should provide a better quantification of etoposide vs control Top2-CCs maps. How many DSB peaks are found in both samples? How many new DSB peaks appear in the etoposide sample? Is there an enrichment of etoposide-induced DSBs at the TSS/gene body of actively transcribed genes?
- 11) The last paragraph of the Results section, describing the effect of methylation at CpG sites, is not at all connected to the previous results and logically motivated. The Authors should provide a more explicit explanation of why they looked into this feature.

MINOR CORRECTIONS

- TOP2 and Top2 are used interchangeably throughout the text. Please choose one version.
- Which cells are RPE-1? Please specify.
- Fig. 2c: is there a statistically significant difference between convergent and divergent/tandem genes?
- Fig. 2b: what do the shaded colors represent? Moving average of the signal? Please clarify in the legend.
- Fig. 6b should be placed on the right of 6a, and 6c below 6a.
- Fig. 6c: what do "Similar" and "Dissimilar" refer to? How was similarity calculated? Please clarify in the legend (and Methods).
- Suppl. Fig. 1 and 4: what do HpM and HpB stand for? Please clarify in the legend (also in other figures where this acronym is used).
- Line 342: also BLESS/BLISS use 3' blunting in the same way as END-seq does, therefore refs. 19 and 51 should be added together with 18. In addition, DSBCapture (Lensing et al, Nat Methods 2016) should be cited here as well as in the Introduction.
- Line 388: remove the parenthesis after ref. 58.
- In almost all the figures there are instances of two or more panels with a single label. Please label each individual plot/panel with a letter, even if two plots are closely related.

FINAL STATEMENT

I greatly appreciated that the Authors provide all the datasets used throughout their analyses in the form of well-organized tables. Also, the table summarizing all sequencing data is very useful and easy to navigate.

Reviewer #3 (Remarks to the Author):

The authors have developed a method (CC-seq) for mapping complexes covalently attached to DNA and the technique might have utility for further studies. It is unclear to me what the breakthrough discoveries of this manuscript are. In general I didn't find it an easy paper to read with a mix of method development, experiments in yeast and human. Furthermore, it wasn't clear to me whether the authors were trying to investigate specific hypothesis-driven questions about TOP2 or were instead just trying to demonstrate utility for the technique. In general I feel there was too much disparate data and it might be better to separate method development and yeast experiments from a paper mapping TOP2 binding in human cells. For me there is too much data and a lack of narrative. Where possible the authors should remove anything extraneous.

Methodology

There are some queries with the methodology which the authors need to clarify - in particular why there is protein attached to the DNA in the absence of VP16 and better validation of SSB vs DSBs. I also feel that as development of CC-seq is a major part of this manuscript the authors should describe the method in more detail. It is not sufficient to reference the Keeney and Kleckner references, particularly as the approach used here is quite different (Keeney purified protein/DNA complexes by CsCl density sedimentation). The methods for CC-seq state that protein/DNA complexes go into the aqueous phase after P/C extraction, but the Keeney and Kleckner PNAS paper say protein/DNA complexes go into the interphase. Please can the authors clarify this.

The authors start by examining Spo11 binding in yeast and demonstrate that CC-seq and Oligo-seq give similar results. In my mind this provides useful validation of the approach. CC-seq also provides very precise mapping of covalent attachment sites and from this data can observe offsets between the strands.

Comments

Fig 1A would be more useful if additional information was added for (i) purification of protein/DNA complexes and (ii) molecular biology steps for processing DNA ends for sequencing.

Fig 1B. Did the authors do a Spo11 W/B to show that these fragments indeed had Spo11 attached to them. It would also be good to include the EtBr stained gel before blotting.

Fig 1C. This graph should not be labelled Spo11-CC as Spo11 have not been specifically enriched. Instead it should just be 'Covalent Complexes'. The same goes for labelling TOP2-CC in Fig 3 and other figures.

Why is there a faint band in Fig 3A -VP16? I am thinking the samples are not extracted stringently enough. See note later about using GnHCl/detergent.

Figure 3B. Instead of discussing the A compartment and B compartment I wonder if it would be better to relate the data to isochores and relate G/C content to Top2 binding. I'm also not quite sure what point the authors are trying to make, particularly as Fig 3C is too simplistic a presentation of global TOP2 binding.

In Figure 3B I am curious why the -VP16 gives a similar pattern to +VP16. Presumably this indicates there are "other" proteins that are covalently attached to the DNA. The slot blots indicate

these proteins are unlikely to be Top2. Do the authors have suggestions what these can be? If the samples are washed with high salt before phenol/chloroform extraction does this go away. In the original Keeney papers sample extraction was harsher using GnHCl with detergent at 65C. Are these proteins covalently attached or just tightly bound? Does the alcohol wash cause proteins to become "precipitated" onto the DNA so they are protected from being extracted by P/C?

In Fig 3B for presenting "TOP2" binding should the authors subtract -VP16 from +VP16?

The authors make an important claim that etoposide induces a majority of TOP2-linked SSBs. They also mention this is consistent with previous studies and suggest it has a sequence component. However, I do not see how, with any certainty, the authors can determine the ratio of ss to ds breaks. Looking at the methods the authors say "SSB sites were defined as those sites without a cognate on the opposite strand at the expected offset of 3 bp." Please can the authors expand on this. Maybe I'm missing something but I don't see how an absence of "read" can convincingly say whether the original break was single stranded or double stranded. Presumably reads are often lost or inefficiently processed and in this case it will appear there has been a single stranded rather than a double stranded break. What do results look like if map ss and ds breaks separately?

Sometimes the authors write VP16 and sometimes write etoposide. Best to be consistent.

Line 235. What is the rationale for focusing on CTCF proximal TOP2 binding sites. Is there something special about CTCF binding sites. Would the results look similar for any transcription factor binding site. What proportion of CC-seq peaks are associated with CTCF binding sites? Likewise what proportion of TOP2 binding sites are found at TSSs. It is difficult to put the data into context.

The data indicates that TOP2 binds to linker (or protein free) DNA (e.g. Fig 4B/4C). In my mind this reflects that TOP2 needs protein free DNA to bind but the implication in the manuscript is that there is some 'special' binding of TOP2 to CTCF sites or at linker DNA at TSS's

I don't see the point of 5B-D. What is the relevance of comparing CTCF to a TSS? They are completely different genomic features. I also don't feel that Fig S5 fits well with the narrative.

I find the result shown in Fig 6D very surprising. Could there be another explanation for this data? Just to clarify is this graph showing C's in a CG context +/- methylation? Essentially all CG's will be methylated so important to distinguish between methylated and unmethylated CG's and not C's in different sequence contexts.

Overview of major changes:

1. Clarified CC-seq methodology in main text and Methods (Lines 94-103; 115-122; 698-721; 733-738) and Fig. 1A, and Fig. S1 (new).
2. Made efforts to more explicitly describe results prior drawing conclusions (e.g. Lines 179-191 and 286-327 - the yeast and human TSS parts).
3. Moved most discussion points out of results sections to clearly separate observations from interpretations.
4. Substantially revised the analysis and comparison of noncognate vs cognate cleavage sites in both yeast and human datasets, including comparisons to a Poisson sampled model, and a randomised dataset (Lines 329-389, and Fig. 6, Fig. S9, Fig. S10).
5. Re-written discussion to be less repetitive, to discuss aspects of data not elaborated elsewhere, and added a small summary figure (Fig. 7).
6. Rearranged panels within supplementary figures so that each Supplementary Figure has a more coherent message and focus.

All text changes relative to original submission are marked in blue.

Reviewers' comments:

Reviewer #1 (Remarks to the Author):

This intriguing paper describes a novel approach ("CC-seq") to mapping covalent protein-DNA linkages in a highly selective and single nucleotide resolution manner. The approach is shown to be generally applicable to mapping of double-strand breaks (DSBs) and (possibly) single-strand breaks (SSBs) made by Spo11 during meiosis and by topoisomerase II (Top2) in response to poisoning by etoposide. The Top2 covalent complexes are mapped in yeast and in cultured human cells. The approach is very clever and appears to be a substantial improvement over existing methods. The data show clearly that the authors have succeeded in developing a technically straightforward, reproducible, specific, and highly sensitive method for enriching and sequencing Spo11 and Top2 covalent complexes. The analysis of the data mostly confirm and extend known patterns, but there are some surprising and interesting new observations that emerge from the high spatial resolution of the data. The datasets will be a valuable resource, but more importantly, the CC-seq method is likely to be powerful new approach to studying this important class of DNA lesion. As such, this manuscript is likely to be of considerable general interest.

Although the paper is mostly well written, there are a number of areas where clarity could be improved or where interpretations could be strengthened. These are all easily addressable by changes to text and figures, with fairly minimal additional data analysis needed. No further experimentation is necessary.

Major points

1. Fig 1A: it would be helpful to provide a cartoon to show how the sequencing library is prepared, especially to allow the reader to understand how DSB and SSB fragments are recovered and sequenced. It would also be helpful to provide more explicit description of the library preparation in the Methods rather than simply referring to the NEB kit used.

We thank the referee for this feedback. We have substantially revised Fig. 1A, and included a new detailed supplementary Fig. S1 to explain how Top2 CC-SSB and Top2 CC-DSB are recovered. We have added methodological details within the opening section of main text, and within the Methods section to help describe the method (Lines 94-103; 115-122; 698-721; 733-738).

2. Line 118: The existence of a base composition bias per se is not evidence of specificity, because library generation or sequencing artifacts would also be expected to have non-random base composition. The pattern needs to be compared to known Spo11 patterns.

We agree that this is an important point and have amended the text as suggested (Lines 137-138).

3. Lines 152-154: The discussion of the positive correlation of Spo11 activity with gene expression could be developed better. Studies that have probed directly for a causal link have failed to demonstrate that transcription levels dictate DSB levels (e.g., PMIDs 1568254, 26245832). Also, Fig S1D is cited here but S1E is meant.

We thank the referee for highlighting this point. Indeed, the correlation between Spo11 cleavage and downstream gene expression that we had identified was very weak and is thus only one of the many factors that contribute to Spo11-DSB activity at a given hotspot. It was not our intention to emphasise this correlation — indeed it was not expected to be visible given prior published work — leading us to wish to test/confirm if this correlation is present in other datasets. Although not presented in our revised manuscript, equivalent analysis using alternative gene expression datasets and Spo11-oligo data reproduced the same weak trend, suggesting it is not an artifact of CC-seq.

Nevertheless, during revision of our manuscript, we adopted coordinates for genomic loci based on recently published transcription start sites of protein-coding genes based on CAGE analysis, rather than gene ATG start coordinates annotated on the Saccharomyces Genome Database. This change reduced the aggregated quantitative trend that is visible in heat maps. Additionally, and more importantly, on a per locus basis we fail to observe any quantitative correlation (scatterplots). Consequently, we have revised the figures (Fig. S5), adjusted the text to remove reference to this positive correlation, and have added the most recent suggested citation (Lines 189-191).

4. The parts of the manuscript that deal with “DSBs” vs. “SSBs” are difficult to understand and therefore not convincing. One problem is that the text doesn't make a clear distinction between fraction of sites (what is measured) and fraction of breaks. This is a fundamental difference that needs to be more clearly dealt with. One cannot know what fraction of events is SSBs because nicks at “DSB” sites will look like DSBs by this analysis. Example on line 292: “we similarly observe more single-strand Top2 CCs (90.1% and 94.2%, respectively) than double-strand Top2 CCs (9.9% and 5.8%, respectively) following etoposide exposure (Figure 6A and S3C).” This statement is incorrect: The observation here is percent of sites, not percent of CCs. Same concern applies to abstract and other places in the text.

5. More generally, a major problem with the SSB analysis is that operational definitions are never really clearly explained, and the text assumes the interpretation rather than providing a neutral description of the findings first, then providing the interpretation. It would be better to provide a clear explanation of the relevant patterns in the data (namely, that there are many locations that show one-sided Top2CC mapping signal), then to give a discussion of interpretation and alternative possibilities. It will be important to explain how one can exclude systematic biases in recovery, library-prep, or sequencing as the source of the one-sidedness. It will also be useful to explain how the purification and sequencing methodology would be expected to behave when presented with a SSB-Top2CC as opposed to a DSB-Top2CC (see also point 1 above).

We thank the referee for these two comments, and for permitting us with an opportunity to substantially revise this section (Lines 329-389, and Fig. 6, Fig. S9, Fig. S10). We agree that the original analysis was limited and not well explained or developed. In our revised manuscript we have attempted to first present an unbiased description of the observations. Additionally, we have made it clear in the opening sections of the manuscript (Lines 119-122) that CC-seq is capable of mapping both DSB ends and SSB-ends that have been converted to a DSB end by sonication.

In the revised later section (Lines 329-389), we describe the presence of sites with significant disparity, that we term, ‘noncognates’, compared to sites with relative parity, that we term ‘cognates’. In order to define these sites we used a more sophisticated method based on the Poisson test, similar to the process used in Pan et al. Cell 2011, and also

compare observations to a randomised dataset, and a simulation based on sampling from a theoretical dataset where 100% of sites are cognate. We then determined that despite inherent fine-scale differences in locations, cognate and noncognate sites are broadly distributed across the yeast and human genome in the same locations, suggesting that they are both measures of Top2 activity. We end this section describing the relative differences in fine-scale DNA sequence composition around cognate and noncognate sites. It is from our interpretation of these differences (namely the relative preference for a cytosine 5' to the scissile phosphodiester bond) that leads to our conclusion that cognate and noncognate sites are likely to represent preferred locations of Top2-CC-DSB and Top2-CC-SSB formation, respectively. Finally, we acknowledge in the discussion that because we are unable to distinguish SSBs from DSBs at cognate sites, that the latter may be stochastic sites of SSB formation (but where there is no strand bias). We present this concept in the discussion because it is rather speculative, and because we have no simple way of testing this hypothesis further using existing methodology.

6. Fig 6D and lines 322-327: The analysis of methylation was completely uninterpretable as presented, and the conclusions are not convincing. Presentation: Neither the text nor the figure explain how methylated sites are defined. One has to dig into the methods to find that methylation has not been assayed directly, but rather that methylation data from a separate study were used. Better presentation is needed. Interpretation: to make the conclusion the authors wish to make, they would need to directly assess methylation on Top2-CC fragments. Since that would probably be prohibitively difficult at this point, the use of existing data is warranted, but more care is needed in interpreting the findings, and alternative interpretations should be considered. For example, individual CpG dinucleotides are not always methylated in all cells in a population, so if methylation within the Top2 binding site is the relevant factor, then cells that lack methylation should be competent to make breaks there. Analysis of methylation fraction vs. Top2-CC frequency should be carried out. Also, the authors interpret their findings solely in terms of the effect of the methyl group on Top2 DNA cleavage, but alternative possibilities include occlusion by nucleosomes or other DNA binding proteins; heterochromatin formation and suppression of transcription; possibility that methylated cytosines tend to be within sequence contexts that are poor substrates for Top2; artifacts of Top2-CC recovery or library preparation. Experimental tests of these ideas are possible (e.g., CC-seq in mutants defective for methylation or heterochromatin formation) but are outside the scope of what is necessary for this paper. But absent such data, there are too many caveats, and conclusions that can be drawn from this analysis are limited.

After reviewing the helpful comments of the referees (see referee #2 additional remarks number 11) we agree that in its current form this analysis is rather limited, and not in keeping with the general message of the paper. We have decided to remove this section in order to develop it further within the scope of future work.

7. The discussion is somewhat repetitive with points made in the introduction and results. The first three paragraphs in particular provide very little new information. It would greatly strengthen the paper if the discussion section were either shortened substantially to remove the redundancy, or rewritten to replace the redundant information with elaboration of points not already covered in detail.

In line with these comments, we have revised our Results and Discussion with an aim to improve clarity, remove repetition and redundancy, and to help to clarify the goals and findings of our study.

Minor points

8. Please number the pages

Page numbers are now included.

9. Fig S1A,B: use consistent labeling (10,100, etc. vs. powers of ten)

This has been corrected (now Fig. S2a,b).

10. S1C: text labels are too small

These panels and text have been resized in revised panel Fig. S2c.

11. Fig. 1E: define “dyad axis”

This has been defined in revised panel Fig. S2d,e.

12. Throughout: the text and figure switch back and forth between “etoposide” and “VP16”. It would be better to pick one and stick with it.

Replaced all cases with “VP16”.

13. Fig. 2C Top: it is unclear why the axis is labeled “density” here since the units are still HpM. Is it always “density” but only sometimes stated as such, or is there something special about the way this graph’s values are calculated?

As the referee notes, we always report density, but were only sometimes explicitly stating it as such. We have removed the term ‘density’ throughout the figures.

14. Fig 3 axis labels are confusing. Some use HpM, others use HpB, and another uses HpBbp. Since the latter two are not defined in the legend, it was hard to understand what was being presented. More importantly, the switch between units seems arbitrary without an explanation for why it needs to be different in different figures.

HpM and HpB are now defined in legends. The inclusion of both (Hits per Million mapped reads per bp or Hits per Billion mapped reads per bp) are frequently used simply to avoid very small numbers in the human datasets where coverage per bp per million reads is very low due to the much larger genome size than *S. cerevisiae*.

15. Fig S5A: what does the lower case “f” mean in “0% FCSf”?

This typo has been corrected.

16. Fig S5C: the micrographs can be made larger since this is a supplemental figure

Enlargement has been made - now Fig. S7c

17. Fig S5E: There is room to write out “TOP2B-/-” so no need to introduce a new (undefined) abbreviation.

Amended - now Fig. S7e

18. Fig 4A top: What do the red and blue triangles signify? Presumably the motifs, and possibly strand orientation? Please specify in the legend.

Amended (Lines 1023-1024)

19. Fig 6B: Y-axis labels are too small

This figure has been revised and the panel enlarged - now Fig. 6e

20. Fig 6B and line 309: The base composition bias is referred to as a “motif” but it is not clear that this term applies here. Was a motif-finding algorithm applied?

Removed all instances of “motif” in relation to the description of the nucleotide skews. We have substantially revised this section of the manuscript (Lines 363-389).

21. It is confusing that the first paragraph under the heading on line 289 is about SSBs vs DSBs, which is not mentioned at all in the heading.

Restructuring of our results section has corrected this problem.

22. Line 339: Nucleotide-resolution mapping of unresected Spo11 DSBs was also reported using S1-seq in yeast (PMID: 29523237, 28059759). This could be cited.

We have added the 2018 reference (Line 397). Many thanks for bringing this Methods paper to our attention.

23. Line 395: DNA bending by Top2 is well established, so the text here gives a misleading impression that the current findings and the sole cited paper are the first to demonstrate it. See, e.g., PMID: 23580548, 25142513.

We have revised this section of the discussion and added these references (Line 468).

24. Lines 401-402: "thus directly demonstrates for the first time that loss of Mre11 activity increases cellular levels of Top2 CCs in *S. cerevisiae*." While formally true as written, this sentence may be misleading because it is likely to be interpreted that this is the first demonstration in any organism, not just the first demonstration in yeast. It would be better to make it explicit that this has been shown already for cultured human cells (Hoa et al., ref. 32 in this ms.) and *Xenopus* extracts (PMID: 26880199, not cited).

We have revised this section of the results (Line 157-158) to describe this observation only in relation to our presented VP16-induced sensitivity (Fig. 2a). We have moved the comparison to known observations in other organisms to the discussion (Lines 481-485).

Reviewer #2 (Remarks to the Author):

MAJOR REMARKS

1) Details of the method. In the first paragraph of the Results section and in Fig. 1a, the Authors describe the CC-seq method which they have developed, by adapting to next-generation sequencing (NGS) a previous biochemical strategy used for enriching Spo11 covalent complexes on silica columns. Unfortunately, the description of the method is minimal to an extent that I had to immediately go to the Methods section to find an explanation, and still did not manage to fully understand how the procedure works:

We greatly appreciate this detailed feedback regarding lack of clarity of our method, which was also highlighted by referees #1 and #3. We have updated Figure 1A to clarify the CC enrichment steps, have added a new supplementary figure (Figure S1) which details how the library preparation works for both Top2-DSBs and Top2-SSBs, and added additional information to the text to clarify these points (Lines 94-103; 115-122; 698-721; 733-738).

--The first step of the protocol is a standard phenol-chloroform DNA purification procedure, although the initial step in ethanol is not conventional (to my knowledge). What is the purpose of this step?

The purpose of this ethanol step is to rapidly denature cellular proteins *in situ*, similar to its use as a 'fixative' in cytological preparations (e.g. PMID 24561827). In the case of yeast, we have empirically determined that this step reduces proteolytic degradation caused by release of vacuolar proteases upon cell lysis in SDS. For consistency, we retained this step within the human cell protocol.

--The Authors state that "By minimizing mechanical shearing of the lysate prior to phenol chloroform extraction, peptides that are covalently linked to high molecular weight DNA segregate into the aqueous phase". Can the Authors clarify what exactly the aqueous phase will contain after following the procedure they describe? Since there is not a prior protein digestion step (e.g., by proteinase K) and since the cells are fixed in ethanol, I imagine that most of the DNA is actually in chromatin form (coated with histones and other protein complexes) and not only bound covalently to proteins such as Top2.

Because cells are lysed at 65°C in a strong detergent solution (2% SDS, 0.5 M Tris, 10 mM EDTA), and then extracted carefully with phenol/chloroform, noncovalently-bound proteins will be removed from the aqueous phase. Secondly, column binding is completed in a buffer containing 300 mM NaCl, 0.2% Triton-X100 and 0.1% N-lauroylsarcosine. Such high salt and detergent will impede ionic binding of any contaminating protein to the purified DNA.

Regardless, if some non-covalent protein-DNA complexes were present in the elution, they would behave in the subsequent library prep the same as free DNA molecules with neither end blocked covalently by a peptide. This would result in both ends receiving a P7 adapter during the first ligation step, which would result in non-amplifiable and non-sequencable molecules. These steps are outlined in Fig. S1c.

We hope that the revisions to the main text, Methods, Fig 1A, and Fig S1, will help to clarify these points.

--After phenol-chloroform, the aqueous phase is passed through a standard miniprep column. In Fig. 1a, the Authors show that this step enriches for CCs between DNA and Top2. How does this work, given the fact that there is no affinity capture? Also, what happens to the other DNA fragments that are presumably coated by proteins (see my comment above), but are not covalently bound to Top2?

We do indeed use Qiagen miniprep columns, but the buffer composition ensures that protein rather than DNA binds to the silica membrane preferentially. Historically, silica membranes have been used to enrich protein rather than DNA (PMIDs: 434459 and 9039264), and they have been more recently adapted for enrichment of DNA, by using different binding buffers including chaotropic salts and alcohol. Indeed, in an earlier publication, the differential binding of protein-DNA complexes vs free DNA to silica membranes in different binding buffers was investigated directly (PMID: 281680). From this work it is clear that column binding in buffer containing 0.3 M NaCl is highly selective for protein-DNA covalent complexes (> 99% bound) and not free DNA (<0.4% bound). We have included this reference in the main text (Line 91).

--A question related to the one above is that even if we assume that the method selects for CCs, it is possible that the DNA end(s) is(are) linked to a protein different than Top2 (possibly, an unknown enzyme). If the Authors are truly able to biochemically isolate protein-bound (covalently) DNA fragments, they should attempt to perform mass spectrometry to elucidate the nature of the proteins/peptides bound.

In principle, it is possible that the CC-seq method described here could be mapping other proteins covalently linked to 5'-DNA termini, provided they were also repairable by Tdp2. We are able to exclude proteins linked to 3'-DNA termini, such as Top1, because the activity of Tdp2 (which has a limited activity on Top1- 3' DNA complexes) produces phosphorylated termini. Only 5'-phosphate termini, but not 3'-phosphate termini, can be directly ligated to the adapters with canonical ends (3'-OH and 5'-P) that we employ.

We have clarified these points in the text (Lines 115-119) and new Fig S1. Furthermore, whilst it is possible that other covalently bound proteins would also lead to enrichment, we believe that we have demonstrated substantial specificity in our maps via controls such as the *spo11-Y135F* mutant, the correspondence of meiotic CC-seq signals to previously published Spo11-oligo seq, the Top2beta KO in human cells, the enrichment in Top2 CC-seq signal caused by etoposide (VP16) treatment in both yeast and human cells, and finally, in yeast cells by the increased CC-seq signal in *sae2D* and *mre11D* mutants, which are known to interfere with Top2-CC DSB repair. As such, whilst we thank referee #2 for the suggestion regarding mass spectrometry experiments, we believe they are beyond the scope of the current study.

--Reading further through the library preparation description in the Methods, I thought I finally understood the trick behind CC-seq, but I am still unsure whether this is truly the case: does it mean that genomic DNA fragments that are not covalently bound to Top2 will be ligated with the P7 adapter on both ends, and thus will not be sequenced, whereas only the fragments bound to Top2, which are end-cleaned by TDP2, can be ligated to the P5 adapter on the end where Top2 was bound, and thus will be sequenced? If this is the case, the Authors should absolutely explain this upfront in the Results section, and also modify Fig. 1a to clearly describe all the steps of the procedure. Also, if this is how CC-seq works, why can't the Authors simply start from purifying DNA over a column followed by the TDP2 step, instead of running through phenol-chloroform?

This is indeed an aspect of our CC-seq method which helps to improve specificity for DNA fragments that are covalently linked to Top2 (or Spo11) at one end. As indicated above, we have substantially revised the text (Line 94-

103 and 115-119), and revised Fig 1A and Fig. S1 to clarify these and other points. However, importantly, as explained above, it is not only the 5'-tyrosine unlinking activity of TDP2 that provides specificity: much of the initial specificity also comes from the bulk removal of free protein (phenol/chloroform extraction), followed by selective enrichment of covalent complexes on silica columns which is used as an affinity step (for CCs) prior to TDP2 treatment.

2) DSBs vs SSBs. In Fig. 6a, the Authors show that most of the signal detected by CC-seq corresponds to SSBs. When I read this I was puzzled, as from my understanding of how the method works I assumed that only DSBs can be captured. Only when reading the corresponding Methods section did I realize that the Authors come to this conclusion based on their analysis of the genome-wide pattern of reads, i.e., they interpret as SSBs those genomic sites where CC-seq yielded only reads on one or the other strand. Unfortunately, I do not think this is a correct interpretation of the data. As the method is based on the NEBNext kit for library preparation, which uses the T4 ligase to ligate the P5 and P7 adapters to the ends of DNA fragments, it is not at all clear how at this step SSBs could be possibly ligated. Furthermore, if SSBs are present within a genomic DNA fragment, they might simply be sealed by the ligase. What I suspect the Authors are seeing in their data could be either of these two DNA structures:

--Single-ended DSBs that have attached Top2 (novel biology?): for example, at stalled replication forks, if the leading strand breaks, and a single-ended DSB is formed, this might be detected by CC-seq (notably, this end does not necessarily have to be bound to Top2, but could be bound to an unknown protein. This brings me to the suggestion I made above to subject DNA fragments with covalently-bound ends to mass spectrometry).

--DSBs induced by Top2, where only one end has remained covalently bound to Top2, or one end is somehow 'masked' by a large protein complex and did not purify (the second hypothesis less plausible).

We thank referee #2 for this feedback, which aligns with comments made by referee #1 (please see referee #1 major remarks 1). We regret that these data were not described nor analysed with sufficient clarity.

We have added a new supplementary Fig. S1b to explain how Top2 CC-SSBs are likely to be converted into sequenceable molecules due to preferential breakage (during sonication) of the ssDNA opposite the pre-existing Top2-linked nick (please refer to PMID: 19795921). As described in our new Fig. S1b and text (Lines 119-122) we expect that Top2 CC-SSBs are converted into one-ended Top2 CC-DSBs upon sonication. These will then be processed into sequenceable molecules during our CC-seq library preparation protocol.

In addition, as suggested by the reviewer, it is also possible that some of the Top2 CC-SSBs we detect are in fact one-ended Top2 CC-DSBs present *in vivo*. However, as discussed above (please see response to referee #1 Major Remark 5), if one side of a Top2 CC-DSB is systematically lost by a biological mechanism independent from Top2 catalysis, such as replication collision, we expect that the average DNA sequence bias at these sites would still remain rotationally symmetrical, and would therefore be indistinguishable from the sequence bias at sites that display no Watson or Crick disparity. Additionally, as described above, the strong denaturants and salt concentrations used during sample preparation and column binding will prevent ionic interactions of other proteins with DNA, and the high degree of etoposide dependency, and specificity of TDP2 in our library preparation for 5' linked phosphotyrosyl ends adds further confidence that any covalent attachment is Top2.

Moreover, it is the fact that sites with disparity ('noncognate' sites) show a pattern broadly similar to 'cognate' sites at all bases except for at the dinucleotides that would have been cleaved were they capable of generating DSBs, that leads us to conclude that noncognate sites really are preferred sites of Top2 CC-SSB formation *in vivo*, and are not the result of technical (or biological) bias that resulted in preferential recovery of just one side of a DSB.

To ensure these points are clear, we have substantially revised the section describing the detection of putative SSBs taking into account our new analyses as described in response to referee #1 above (Lines 329-389, and Fig. 6, Fig. S9, Fig. S10)

The Authors must absolutely clarify these issues and provide a better interpretation of what they name as SSBs are in reality. Towards this goal, it would be useful if the Authors performed an experiment with or without a Top1 inhibitor or another drug causing a dramatic increase in the number of SSBs, and see what CC-seq reveals.

We have substantially revised this section of the manuscript, where we hope to have clarified our analysis and conclusions (Lines 329-389, and Fig. 6, Fig. S9, Fig. S10). Moreover we have tried to present more explicitly in the opening results section how putative CC-SSBs will be converted to mappable CC-DSB ends due to sonication opposite a preexisting nick (Lines 119-122).

Regarding Top1: we certainly plan to expand CC-seq to the selective mapping of Top1 CCs in the future. In its current form, however, the method specifically excludes CCs linked at 3' DNA termini (new Fig. S1b).

ADDITIONAL REMARKS

1) Although the manuscript is well written, I found that, especially in the Results section, in many occasions the results were only superficially explained and many details were left implicit. This is also the case in figure legends and in plot labels, where many details are left unexplained and abbreviations are often not spelled out (see Minor Comments below). The Authors should strive to provide more details and describe the results in a more explicit manner.

Thanks to the referee #2 for this feedback. We have attempted to address this throughout the revised manuscript and figure legends.

2) Related to the previous point, the Results section often contains interpretations/discussions that should be left for the Discussion section only. Also, the tense occasionally switches from past (narrative style, which is easy to follow) to present, which makes it difficult to follow what is a result and what is more a discussion point. An example would be lines 147 to 154: "[...] the fact that *S. cerevisiae* Top2 CC signal is not correlated [...]. By contrast, Spo11 activity is positively correlated [...]."

In line with our other general revisions, we moved most discussion points from the Results to the main Discussion, and have changed the tense to a past narrative style when describing our results.

3) Further related to the previous points, throughout the manuscript I had the feeling that yeast and human results are often mixed. It would be clearer to present all the findings in *S. cerevisiae* first, and then show the results obtained using human cells (the same in the Discussion).

We thank the referee for this comment, and have attempted to accommodate this suggestion. Our manuscript covers three separate biological systems, and we believe where possible we have separately introduced and described the data in the manner suggested (e.g. Fig. 1 is yeast meiosis, Fig. 2 is yeast Top2; Fig. 3-5 are human Top2). However, there are instances where we believe it is relevant to draw comparisons between the systems, such as the sequence bias comparisons of Top2 between yeast and human (Fig. 6), the relationship between Top2 and Spo11 to nucleosome occupancy (Fig. S4) and transcription (Fig. S5), and the interesting differential relationship of Top2 CC-seq with gene length in yeast and human (Fig. S9).

As part of our revisions, we have additionally tried to simplify the number of composite panels within each figure, and added additional labelling to relevant panels. We have spent a long time considering the best way to present our observations with clarity, and hope these changes will prove helpful to the reader.

4) It is not clear why the Authors used a human spike-in to normalize yeast libraries and not, for example, a yeast spike-in to normalize human libraries?

This aspect of the method was actually only developed after the human libraries had been processed and sequenced. In our future experiments we will certainly employ a yeast spike-in in our human experiments.

5) Line 119: while describing the results shown in Fig. 1e, the Authors should specify how the actual Spo11 binding motif looks like: was this previously known? If not, is the observed sequence preference in line with the structure of DNA-bound Spo11 (if available), as shown for Top2 in Fig. 6?

Indeed the Spo11 sequence bias is very similar to that identified from a prior study. As also suggested by Referee #1, we have clarified this point within the text (Line 137-138).

5) Fig. 3d: the Authors should quantify the correlation between Top2 peaks and CTCF and H3K4Me3 and H3K27Ac peaks, and also provide a quantification of the relative distribution of Top2 peaks in different parts of the genome (coding vs non-coding, exons, introns, TSS, etc).

We do not believe that the fine-scale nucleotide-resolution peaks detected by CC-seq maps makes them especially suitable for broad-scale peak calling. Top2 CC-seq signal is not focussed in clear hotspots like Spo11 CC-seq, for example (Fig. 2b), and in general, Top2 CC-seq signals are much more broadly distributed across the genome in both yeast and human than the specific binding sites of factors like CTCF (Fig. 3f), or modified histones (Fig. 3f). As such, we have not undertaken any peak calling of our data, but have rather aggregated CC-seq around sites of interest such as TSSs and CTCF. Once published, researchers will have full access to our raw data, permitting them to undertake in the future any analyses such as those that you suggest.

Additionally, to address the question about the correlation between Top2 and CTCF and marks of active transcription, in our revised manuscript we have expanded the quantitative analysis between CC-seq signal and CTCF binding and transcription via scatterplots with r-value statistics presented in Fig. 4d and Fig. 5e-h.

6) The Authors nicely show a strong correlation between DSB maps obtained by End-seq (indirectly) and CC-seq profiles. Even though the Authors provide a clear rationale for developing CC-seq, these methods might be perceived as redundant, in the sense that, in the end, they all show the distribution of DSBs genome-wide (with END-seq and BLESS/BLISS or DSBCapture also detecting DSBs not originating from topoisomerases). The Authors should better contrast these methods in the Discussion, and highlight the advantages of CC-seq over other DSB-detection methods.

We thank the referee for this feedback, and have better emphasized the utility and non-redundancy of CC-seq when compared to previous DSB mapping methods, such as END-seq. As the referee notes, these techniques may seem redundant, especially considering their spatial correlation at broad scale (Fig. 3b). However, unlike CC-seq, END-seq is not accurate for the site of Top2 cleavage at fine scale (Fig. 3c); unable to recover and map real SSBs (a consequence of *in situ* ligation of adapters prior to sonication); and not specific for, or perhaps even incapable of directly mapping protein-linked DNA ends. We believe that these points make CC-seq a far more suitable and descriptive tool for mapping Top2 activity genome-wide than the methods mentioned. We have emphasised these points and discussed potential reasons for differences between CC-seq and END-seq in the discussion (Lines 446-454).

7) In their CTCF analysis, have the Authors examined whether the loops that have a Top2-CC on their flanks also contain a gene that is actively transcribed? It would be helpful to classify loops based on whether they contain or not a gene (inside the loop or flanking it) and whether the gene is active or not.

We thank the referee for raising this interesting point (related to <https://www.biorxiv.org/content/10.1101/485763v1> Figure 4). However, given the breadth of data already presented in this manuscript, we have chosen to leave such complex bioinformatic analyses to a future study, in which it may be more of a focus.

8) The Authors show that Top2-CC in human cells are abundant within the gene body of transcribed genes, in line with previous findings by other DSB-capturing methods. Can the Authors provide a better quantification of Top2-CCs in gene bodies? Are DSBs occurring all along the gene body or preferentially at the 5' end? Is there a preference for exons, introns or exon-intron boundaries?

Indeed, the CC-seq signal is preferentially associated within the first few kilobases of the transcribed genes. We refer the reviewer to Fig. 5 and Fig. S9 where genes are stratified by either expression level or gene length, and a 5 kb window is plotted. In both cases the enrichment of CCseq is in the first few kb downstream of the transcription start site (TSS). We have not investigated exon-intron boundaries in great detail. However, although there is a minor pattern in CC-seq maps at exon-intron junctions, there is also a very high sequence skew due to the splicing signal sequence, which we are concerned may be generating the CC-seq patterns due to the influence primary DNA sequence has on Top2 catalysis. Because of this complexity, and in the interest of space, we would prefer not to present this result, which we feel would take much further analysis and experimentation to correctly understand. Published datasets can, however, be freely explored by others in order to ask any questions.

9) Lines 268-269: how did the Authors come to the conclusion that "[...] Top2 activity is not proportional to local RNA polymerase activity when assayed at finer scale". Figure 5a actually shows the opposite. Where is the data supporting the Authors' statement?

We apologise for the confusion arising from our original text, which was referring to the correlative analysis performed by others on END-seq relative to transcription (GRO-seq) in Canela et al. Cell 2017. To avoid such confusion, we have moved such comparative comments to the Discussion (e.g. Lines 446-454). Additionally, we have taken the opportunity to further investigate the quantitative and spatial relationships between CC-seq maps, transcription measured by GRO-seq, and nucleosome occupancy (Fig. 5g-j). These analyses revealed striking patterns of Top2 activity in the regions within and immediately adjacent to promoters (Fig. 5i,j). We are excited by these observations and hope these additional panels will be of interest to readers.

10) Related to point 5), the Authors should provide a better quantification of etoposide vs control Top2-CCs maps. How many DSB peaks are found in both samples? How many new DSB peaks appear in the etoposide sample? Is there an enrichment of etoposide-induced DSBs at the TSS/gene body of actively transcribed genes?

As discussed above, we don't believe that the qualitative nature of the CC-seq maps (millions of low frequency hits) makes them suited for such comparative peak-calling analyses, which even if successful, are likely to reveal enriched signal only in a very small set of peaks (containing a tiny fraction of the total CC-seq dataset), and which are thus unlikely to be representative of the broader genome-wide pattern. Moreover, as clarified in our revised manuscript (Lines 209-212; 271-273) we believe that untreated libraries contain a low level of CC enrichment, due to a low level of spontaneous Top2-CC formation in cells. To emphasise this point, we have provided an overlay of the CTCF-proximal signal, where the relatively focal Top2-CC signal enables detection of a periodic pattern that is near-identical to that present VP16-treated cells, but with much lower amplitude due, we assume, to a much higher background level of reads mapping non-specifically across the genome.

11) The last paragraph of the Results section, describing the effect of methylation at CpG sites, is not at all connected to the previous results and logically motivated. The Authors should provide a more explicit explanation of why they looked into this feature.

In response to combined referee comments, we have removed this analysis from the manuscript such that it can be developed in greater detail in a future study.

MINOR CORRECTIONS

--TOP2 and Top2 are used interchangeably throughout the text. Please choose one version.

We thank the referee for this suggestion. This is actually intentional, as “Top2” refers to the yeast protein, whereas “TOP2” refers to the human protein. Although we agree that this is perhaps an unimportant distinction here, we wish to maintain this for clarity. We use “Top2” when the sentence is ambiguous.

--Which cells are RPE-1? Please specify.

We have added this information to the Results (Line 196-197) and Methods (Line 565)

--Fig. 2c: is there a statistically significant difference between convergent and divergent/tandem genes?

The Kruskal-Wallis one-way analysis of variance (ANOVA) across all groups showed significance (p-value < 2.2×10^{-16}). Subsequently, all unpaired two-sample Wilcoxon tests between each group on Figure 2C (including between Top2 Tandem and Top2 Divergent, where the effect size is very small) showed significance (p-values < 0.001). We have updated the figure legend (now Fig. 1d) to clarify this.

--Fig. 2b: what do the shaded colors represent? Moving average of the signal? Please clarify in the legend.

Yes, they are smoothed average curves. We have amended figure legends accordingly.

--Fig. 6b should be placed on the right of 6a, and 6c below 6a.

This figure has been extensively revised following a more developed description and statistical analysis of the data (Lines 329-389, and Fig. 6, Fig. S9, Fig. S10).

--Fig. 6c: what do “Similar” and “Dissimilar” refer to? How was similarity calculated? Please clarify in the legend (and Methods).

These terms have been replaced by “Log2 deviations” with a colour scale (Fig. 6f), with clarification in the figure legend (Line 1077-1083) and Methods (Line 954-957).

--Suppl. Fig. 1 and 4: what do HpM and HpB stand for? Please clarify in the legend (also in other figures where this acronym is used).

These terms have been added to each figure legend. (Hits per Million/Billion mapped reads per base pair).

--Line 342: also BLESS/BLISS use 3' blunting in the same way as END-seq does, therefore refs. 19 and 51 should be added together with 18. In addition, DSBCapture (Lensing et al, Nat Methods 2016) should be cited here as well as in the Introduction.

We thank the referee for highlighting this point about blunting potentially affecting BLISS/BLESS as well as END-seq. We have added DSBCapture reference to the discussion (Line 401). In the introduction we focus only on those pre-existing methods that have been used to map VP16-induced DSBs.

--Line 388: remove the parenthesis after ref. 58.

We have corrected this.

--In almost all the figures there are instances of two or more panels with a single label. Please label each individual plot/panel with a letter, even if two plots are closely related.

We have modified figure labelling as requested.

FINAL STATEMENT

I greatly appreciate that the Authors provide all the datasets used throughout their analyses in the form of well-organized tables. Also, the table summarizing all sequencing data is very useful and easy to navigate.

We thank the referee for these comments.

Reviewer #3 (Remarks to the Author):

The authors have developed a method (CC-seq) for mapping complexes covalently attached to DNA and the technique might have utility for further studies. It is unclear to me what the breakthrough discoveries of this manuscript are. In general I didn't find it an easy paper to read with a mix of method development, experiments in yeast and human. Furthermore, it wasn't clear to me whether the authors were trying to investigate specific hypothesis-driven questions about TOP2 or were instead just trying to demonstrate utility for the technique. In general I feel there was too much disparate data and it might be better to separate method development and yeast experiments from a paper mapping TOP2 binding in human cells. For me there is too much data and a lack of narrative. Where possible the authors should remove anything extraneous.

We appreciate this general feedback from referee #3. In our revised manuscript we have focussed on clarifying the novel methodology (Lines 94-103; 115-122; 698-721; 733-738 and Fig. 1A, Fig. S1), comparing our approach better with preceding techniques (Lines 397-405 in Discussion), and sharpening our focus on novel aspects of the biology (Yeast Top2 activity within IGRs, Fig. 2; broad and finescale connections of CC-seq to transcription in human cells, Fig. 5; and evidence for Top2 SSB formation, Fig 6). We have additionally summarised our findings in a simplified accessible final Discussion figure (Fig. 7).

Methodology

There are some queries with the methodology which the authors need to clarify - in particular why there is protein attached to the DNA in the absence of VP16 and better validation of SSB vs DSBs. I also feel that as development of CC-seq is a major part of this manuscript the authors should describe the method in more detail. It is not sufficient to reference the Keeney and Kleckner references, particularly as the approach used here is quite different (Keeney purified protein/DNA complexes by CsCl density sedimentation). The methods for CC-seq state that protein/DNA complexes go into the aqueous phase after P/C extraction, but the Keeney and Kleckner PNAS paper say protein/DNA complexes go into the interphase. Please can the authors clarify this.

We thank the referee for this feedback, and have taken steps to emphasise and clarify our novel methodology including additional primary references to silica-based CC-enrichment (Lines 94-103; 115-122; 698-721; 733-738 and Fig. 1A, Fig. S1).

With regards to the apparent discrepancy with a previous report of CC isolation from the interphase, we note that prior observations were made following restriction digestion of protein-linked DNA molecules into fragments ~5 kb in size. By contrast we specifically avoid mechanical shearing of DNA prior to phenol-chloroform extraction, and estimate the average DNA size at the time of phenol extraction to be >50 kb. Importantly, despite the work presented by Keeney and Kleckner (1995, 1997), we are able to empirically determine that covalent complexes are enriched in the aqueous phase, because we detect Top2 in these DNA preparations immunologically (Fig. 3a), and Spo11-DNA

molecules by Southern blotting (Fig. 1b). We hypothesise that the relative partitioning of protein-DNA CCs between the organic, inter, and aqueous phases is dependent on the relative molecular weights of the peptide and nucleic acid moieties. Thus, by employing careful handling—in order to minimize shearing of high molecular weight DNA—we ensure that protein-DNA CCs partition to the aqueous phase. We have clarified this point in the text (Lines 98-100), and by redesigning Fig. 1a.

The authors start by examining Spo11 binding in yeast and demonstrate that CC-seq and Oligo-seq give similar results. In my mind this provides useful validation of the approach. CC-seq also provides very precise mapping of covalent attachment sites and from this data can observe offsets between the strands.

Comments

Fig 1A would be more useful if addition information was added for (i) purification of protein/DNA complexes and (ii) molecular biology steps for processing DNA ends for sequencing.

We have substantially revised Fig. 1a to provide this information, and provided additional library preparation details in Fig. S1.

Fig 1B. Did the authors do a Spo11 W/B to show that these fragments indeed had Spo11 attached to them. It would also be good to include the EtBr stained gel before blotting.

Unfortunately protein-DNA CCs do not migrate normally during electrophoresis. This actually prevents their observation by Southern blotting unless samples are pre-incubated with proteinase K. We provide a control in which the sample was not pre-incubated with proteinase K (Fig. 1B, lane 1). In this lane, it is clear that the bands at the expected size for the Spo11-DSB fragments are specifically depleted, relative to in the lane where the sample was preincubated with proteinase K (lane 2). We have revised the labelling of this figure to make it clear that all lanes except lane 1 had prior proteinase K incubation. Unfortunately, we do not have an ethidium stained image to pair with the presented Southern blot. In other examples, digested DNA is visible in input lanes 1-3, with very little DNA visible in washes (lanes 4-5), or the elution (lane 6), consistent with >99% of the parental non-CC DNA fragments being removed (as shown in lane 6).

Fig 1C. This graph should not be labelled Spo11-CC as Spo11 have not been specifically enriched. Instead it should just be 'Covalent Complexes'. The same goes for labelling TOP2-CC in Fig 3 and other figures.

We have revised all such figure labelling in line with this suggestion.

Why is there a faint band in Fig 3A -VP16? I am thinking the samples are not extracted stringently enough. See note later about about using GnHCl/detergent.

We believe that this very faint band (< 0.2% of the intensity of band in etoposide-treated sample) is reflective of Top2 CCs which are detectable even in the absence stabilization by etoposide. In support of this, we note that in untreated samples the CTCF aggregation analyses, where the CC-seq signal is relatively dense, shows similar, yet far weaker, signal patterns to the etoposide-treated sample (Fig. 4e). In further support of this, the broad-scale distribution of CCs in -VP16 samples is similar to that in +VP16 samples (Fig. 3d,e).

Figure 3B. Instead of discussing the A compartment and B compartment I wonder if it would be better to relate the data to isochores and relate G/C content to Top2 binding. I'm also not quite sure what point the authors are trying to make, particularly as Fig 3C is too simplistic a presentation of global TOP2 binding.

Our understanding of the literature around this subject is that the five principle isochore classes (L1, L2, H1, H2 and H3) can be broadly divided into two main classes: gene-rich/GC-rich genome core (H2, H3) and gene-poor/GC-poor

gene desert (L1, L2, H1), and that these two classes correspond to chromatin compartments A and B, respectively (please refer to PMID 28060840, 26619076). As such, we suggest that correlating Top2 activity with either chromatin compartments or with isochores/ GC-content would produce the same conclusion - that Top2 activity is associated with active, gene-rich chromatin. The aim of Fig. 3b (now Fig. 3d), is therefore to demonstrate a very zoomed-out broadscale view of the patterns of Top2 activity relative to broadscale patterns of supercoiling and Hi-C compartment classification in the human genome. We think this overview is a valuable inclusion to the paper, from where we then zoom in to a smaller region (Fig. 3f), and then zoom in further to specific genomic features (e.g. Fig. 4 and Fig. 5).

In Figure 3B I am curious why the -VP16 gives a similar pattern to +VP16. Presumably this indicates there are “other” proteins that are covalently attached to the DNA. The slot blots indicate these proteins are unlikely to be Top2. Do the authors have suggestions what these can be? If the samples are washed with high salt before phenol/chloroform extraction does this go away. In the original Keeney papers sample extraction was harsher using GnHCl with detergent at 65C. Are these proteins covalently attached or just tightly bound? Does the alcohol wash cause proteins to become “precipitated” onto the DNA so they are protected from being extracted by P/C?

As noted above, we do believe that the similar broad-scale spatial distribution of CCs in -VP16 and +VP16 samples (revised Fig. 3d) is due to detection of Top2 CCs in both unperturbed and Top2-poisoned conditions (Please see response above for further evidence). The referee comments that the faintness of the anti-Top2 slot blot band in the -VP16 sample suggests that we may instead be detecting DNA attached to other proteins in our CC-seq libraries - VP16. We would like to offer that despite lower absolute yield of CC-seq conducted on -VP16 samples, signal-to-noise remains high enough to detect real Top2 CCs which colocalize (at broad-scale) with those detected in +VP16 samples, where both the yield and the signal-to-noise are significantly higher. Moreover, where CC-seq signal is relatively dense (such as CTCF sites), we still observe similar, yet far weaker, signal patterns to the etoposide-treated sample (Fig. 4e).

Referee #3 and #2 both suggest that we may be insufficiently removing non-covalent proteins from DNA, and that this may be causing signal artifacts. We would additionally please like to refer referee #3 to our response to referee #2, where we have responded to this point in detail:

Because cells are lysed at 65°C in a strong detergent solution (2% SDS, 0.5 M Tris, 10 mM EDTA), and then extracted carefully with phenol/chloroform, noncovalently-bound proteins will be removed from the aqueous phase. Secondly, column binding is completed in a buffer containing 300 mM NaCl, 0.2% Triton-X100 and 0.1% N-lauroylsarcosine. Such high salt and detergent will impede ionic binding of any contaminating protein to the purified DNA.

Regardless, if some non-covalent protein-DNA complexes were present in the elution, they would behave in the subsequent library prep the same as free DNA molecules with neither end blocked covalently by a peptide. This would result in both ends receiving a P7 adapter during the first ligation step, which would result in non-amplifiable and non-sequenceable molecules. These steps are outlined in Fig. S1c.

We hope that the revisions to the text and figures will help to clarify these points (Lines 94-103; 115-122; 698-721; 733-738 and Fig. 1A, Fig. S1).

In Fig 3B for presenting “TOP2” binding should the authors subtract -VP16 from +VP16?

As discussed in our responses above, and as described in the revised manuscript (Lines 209-212; 271-273), we think that the signal observed in -VP16 is reflective of a low level of spontaneous Top2 CCs present in unchallenged cells (please see above), and therefore we don't think that it is appropriate to subtract this from the signal observed in +VP16.

The authors make an important claim that etoposide induces a majority of TOP2-linked SSBs. They also mention this is consistent with previous studies and suggest it has a sequence component. However, I do not see how, with any

certainty, the authors can determine the ratio of ss to ds breaks. Looking at the methods the authors say “SSB sites were defined as those sites without a cognate on the opposite strand at the expected offset of 3 bp.” Please can the authors expand on this. Maybe I’m missing something but I don’t see how an absence of “read” can convincingly say whether the original break was single stranded or double stranded. Presumably reads are often lost or inefficiently processed and in this case it will appear there has been a single stranded rather than a double stranded break. What do results look like if map ss and ds breaks separately?

We thank referee #3 for this feedback, which is in alignment with comments made by referee #1 and #2. We present a similar response here, as we believe that it addresses the points raised here by referee #3:

We thank the referee for these comments, and for permitting us with an opportunity to substantially revise this section. We agree that the original analysis was limited and not well explained or developed. In our revised manuscript we have attempted to first present an unbiased description of the observations. Specifically, we have made it clear in the opening sections (Lines 119-122) and in a new supplementary figure (Fig. S1b) that CC-seq is capable of mapping from DSB ends and SSB-ends that are converted to a DSB end by sonication (please refer to PMID: 19795921).

Secondly, in the revised later section (Lines 329-389, and Fig. 6, Fig. S9, Fig. S10), we describe the presence of sites with significant disparity, that we term, ‘noncognates’, compared to sites with relative parity, that we term ‘cognates’. In order to define these sites, we used a more sophisticated method based on the Poisson test, similar to the process used in Pan et al. Cell 2011, and also compare observations to a randomised dataset, and a simulation based on sampling from a theoretical dataset where 100% of sites are cognate. We then determined that despite inherent fine-scale differences in locations, cognate and noncognate sites are broadly distributed across the yeast and human genome in the same locations, suggesting that they are both measures of Top2 activity. We end this section describing the relative differences in fine-scale DNA sequence composition around cognate and noncognate sites. It is from our interpretation of these differences (namely the relative preference for a cytosine 5’ to the scissile phosphodiester bond) that leads to our conclusion that cognate and noncognate sites are likely to represent preferred locations of Top2 CC-DSB and Top2 CC-SSB formation, respectively. Finally, we acknowledge in the discussion that because we are unable to distinguish SSBs from DSBs at cognate sites, that the latter may be stochastic sites of SSB formation (but where there is no strand bias). We present this concept the discussion because it is rather speculative, and because we have no simple way of testing this hypothesis further using existing methodology.

Sometimes the authors write VP16 and sometimes write etoposide. Best to be consistent.

We have changed all instances to VP16, and first defined this as the drug etoposide on Line 58.

Line 235. What is the rationale for focusing on CTCF proximal TOP2 binding sites. Is there something special about CTCF binding sites. Would the results look similar for any transcription factor binding site. What proportion of CC-seq peaks are associated with CTCF binding sites? Likewise what proportion of TOP2 binding sites are found at TSSs. It is difficult to put the data into context.

These experiments were motivated by recent studies (e.g. Canela et al. 2017) where significant VP16-induced DSB signal was detected at CTCF sites. Thus, we wished to determine whether CC-seq, which directly maps TOP2 covalent complexes not DSB ends, also displays this pattern. We have clarified this in the revised manuscript (Lines 278-283; 435-444). Since our manuscript is already rather extensive, we have not at this stage compared or included CC-seq maps at other transcription factor binding sites. Rather, once published, our raw datasets will be available for researchers to use freely to ask such questions.

In regards to the comparison between TSS-proximal and CTCF-proximal CC-seq signal, we would like to refer referee #3 to Fig. S13d-e, where we directly compare the degree of enrichment found around these two distinct types of loci, and furthermore contrast our findings with those data present in END-seq maps.

The data indicates that TOP2 binds to linker (or protein free) DNA (e.g. Fig 4B/4C). In my mind this reflects that TOP2 needs protein free DNA to bind but the implication in the manuscript is that there is some 'special' binding of TOP2 to CTCF sites or at linker DNA at TSS's

Indeed, our global analyses (Fig. 4e and Fig. S4a,b), clearly indicate an anticorrelation between CC-seq signal and nucleosome occupancy. Nevertheless, simple visual inspection (Fig. 3f), and subsequent quantitative analysis (Fig. 4 and Fig. 5) argue that CTCF and TSS sites are two genomic features with increased TOP2 activity. Moreover, we have extended our analysis of TSS-proximal CC-seq data, comparing to both GRO-seq and nucleosome occupancy. From this analysis, it is clear that Top2 signal does not only track with increasing nucleosome depletion across the TSS, but in fact appears suppressed towards the region that is most depleted of nucleosomes (Fig. 5i) - perhaps due to occlusion by other factors, or due to a lack of topological stress in this area. Moreover, within the core promoter, we observe a distinct peak of Top2 CC-seq directly over the location of promoter proximal pausing by RNAPoIII (Fig. 5j). We hope that in our revised manuscript we have made these points clear (Lines 164-191; 304-327; 435-444; 456-463).

I don't see the point of 5B-D. What is the relevance of comparing CTCF to a TSS? They are completely different genomic features. I also don't feel that Fig S5 fits well with the narrative.

Our motivation was as a counterpoint to the recent paper (Canela et al 2017), which largely focuses VP16-induced DSBs around CTCF sites, and failed to find a connection to transcription, something that our data disagree with. To emphasise this point, we have revised the presentation of these data into a single comparative figure of END-Seq vs CC-seq (Fig. S13), and have described this comparison (and possible explanations for the difference) in the revised discussion (Lines 446-454). The point of the overlay and comparisons here (Fig. S13d,e) is to compare the relative level of enrichment of CTCF-proximal vs TSS-proximal sites in CC-seq vs END-seq maps.

I find the result shown in Fig 6D very surprising. Could there be another explanation for this data? Just to clarify is this graph showing C's in a CG context +/- methylation? Essentially all CG's will be methylated so important to distinguish between methylated and unmethylated CG's and not C's in different sequence contexts.

Whilst we also consider this observation intriguing, in light of additional comments from reviewers, we have decided to remove this analysis such that it can be developed more completely in the future.

REVIEWERS' COMMENTS:

Reviewer #2 (Remarks to the Author):

I think the Authors have done a great job in revising their original manuscript following all the three Reviewers' suggestions, and they have satisfactorily addressed all my previous concerns.

In particular, I think the new Fig. 1a and Supplementary Fig. 1 explain the method in a much better manner than before, and have clarified all the questions on the CC enrichment procedure, which I had raised.

Furthermore, the analysis of CC patterns around human CTCF sites and TSSs is now much stronger and very compelling. Overall, the quality of all the figures is enhanced in this revised version, and the clarity of the main text has greatly improved.

Reviewer #3 (Remarks to the Author):

I am satisfied with the authors' responses. I think this is an exciting but challenging area of research. The paper provides a useful new methodology and new ideas that can be further explored by the field.